# Beyond NL2Code: A Structured Survey of Multimodal Code Intelligence

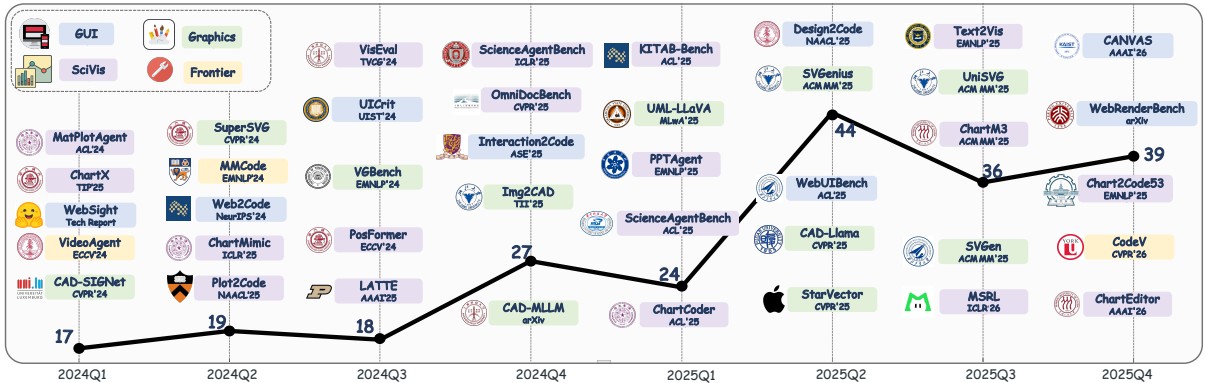

Figure 1: Quarterly publication trend for the surveyed multimodal code intelligence literature.

## Abstract

While Large Language Models (LLMs) have substantially advanced text-to-code synthesis, many real programming tasks specify intent through visual artifacts such as screenshots, charts, documents, vector drawings, videos, and interactive states. These tasks require models to connect visual perception with executable programs, because correctness depends not only on syntax, but also on layout, geometry, data semantics, editability, interaction behavior, and domain-specific constraints after execution. This survey examines Multimodal Code Intelligence, covering systems that generate, edit, refine, execute, or reason with code under visually grounded inputs and outputs. We first formulate the field by the role that code plays in each task, distinguishing code as a rendered artifact, an editable symbolic structure, a scientific representation, an intermediate reasoning trace, or an executable policy or tool interface. We then organize benchmarks and methods into four domains: Graphical User Interface, Scientific Visualization, Structured Graphics, and Frontier Tasks and Frameworks. This taxonomy connects mature artifact-generation problems with emerging agentic and unified settings, and allows us to compare what different tasks treat as evidence of correctness. Across the literature, visual similarity remains useful but incomplete; reliable evaluation also requires evidence about semantics, execution, interaction, transfer, and failure attribution. Looking ahead, we argue that future research may benefit from four verification-centered directions. Multi-signal validation can combine complementary correctness evidence, multi-state verification can test behavior over execution trajectories, cross-task transfer testing can probe reusable visual-code skills, and verifiable agent traces can expose whether agent actions are grounded in visual evidence. Together, these directions may move multimodal code generation from single-output imitation toward evidence-grounded executable systems. An ongoing, dynamically updated project and resources associated with this survey have been released at https://anonymous.4open.science/r/Awesome-Multimodal-LLM-for-Code-2031.

# Contents

# 1 Introduction

Code provides a formal interface between high-level human intent and executable computation, translating abstract specifications into executable instructions (Sun et al., 2024). Large Language Models (LLMs) have substantially advanced Natural Language-to-Code (NL2Code) generation, where models synthesize executable programs from textual specifications (Chen et al., 2021; Li et al., 2023; Rozière et al., 2024; Zan et al., 2023; Jiang et al., 2024; Zhu et al., 2024b; Yang et al., 2025d). This paradigm requires alignment between linguistic intent and formal program syntax, and it has become a central interface for automating software and tool-use workflows.

The scope of NL2Code now extends beyond function-level synthesis to repository-level engineering, issue resolution, and code-mediated tool use. Function-level benchmarks study standalone code snippets (Austin et al., 2021; Li et al., 2022b; Jain et al., 2024), while repository-level and software-engineering tasks require cross-file dependencies, project-wide coherence, debugging, and repair (Liu et al., 2023c; Zhang et al., 2023; Jimenez et al., 2023; Yang et al., 2024c; Zhang et al., 2024b). Code also functions as an action interface for invoking tools, querying structured resources, and orchestrating agentic workflows (Schick et al., 2023; Gao et al., 2023; Wang et al., 2024d). These capabilities make code useful beyond text completion, but they remain largely text-centered when the task intent is specified visually.

Despite these advances, most NL2Code approaches rely solely on textual descriptions. In practice, visual signals serve as a high-bandwidth and intuitive medium for communication. Unlike sequential text, a single image can efficiently encode dense spatial hierarchies and complex structural information that are challenging to articulate verbally. This modality gap becomes especially critical in visual-centric domains such as frontend development (Si et al., 2025; Laurençon et al., 2024), data visualization (Yang et al., 2024a; Zhao et al., 2025d), and computer-aided design (Wu et al., 2021; 2025c), where the generated code yields fundamentally visual outputs. In these scenarios, relying solely on text to describe intricate user interface layouts or precise geometric structures is both inefficient and prone to information loss, often leading to a misalignment between human intent and the resulting code. To bridge this gap, the recent advent of Multimodal Large Language Models (MLLMs) integrates visual perception with logical reasoning (Liu et al., 2023b; Shen et al., 2025). Spurred by the need to address these real-world bottlenecks, the field of Multimodal Code Intelligence has emerged. This approach enables models to understand visual inputs directly, treating visual perception not as an auxiliary feature, but as a core prerequisite for automating visually-driven programming tasks.

In this survey, we provide a structured overview of recent advancements in Multimodal Code Intelligence. We first establish a formal problem formulation for various multimodal code generation tasks. This formulation connects each domain to the dominant role of code, such as rendered artifact, editable structure, reasoning trace, or executable policy. To categorize the rapidly expanding body of literature, we organize existing research into four organizing domains: (1) Section 3 reviews Graphical User Interface, encompassing the generation of web and mobile applications; (2) Section 4 delves into Scientific Visualization, ranging from statistical charts and structured documents to academic presentations; (3) Section 5 focuses on Structured Graphics, covering Scalable Vector Graphics (SVG), diagrams, and Computer-Aided Design (CAD); and (4) Section 6 explores emerging Frontier Tasks and Frameworks, such as programmatic visual manipulation, video generation, and unified multimodal models. For each domain, we review the landscape of benchmarks and methodologies, as structured in Figure 4 and 5. Each subsection ends with a Scope and Trajectory paragraph that identifies the task objective, dominant evidence signal, and remaining validation gap, and each domain section concludes with a Takeaway paragraph that summarizes the domain-specific bottleneck. Looking forward, Section 7 develops a verification-centered agenda that connects these bottlenecks to multi-signal validation, multi-state verification, cross-task transfer testing, and verifiable agent traces. These directions correspond to validating generated or edited artifacts, tool-use traces, and executable policies after rendering, execution, interaction, or replay.

**Survey Methodology.** We follow a staged review protocol consisting of source collection, candidate screening, taxonomy assignment, and manual consistency checks by the authors. Candidate papers are collected from arXiv and major venues in artificial intelligence, computational linguistics, software engineering, and related fields. The manuscript uses a literature snapshot updated through January 2026, with emphasis

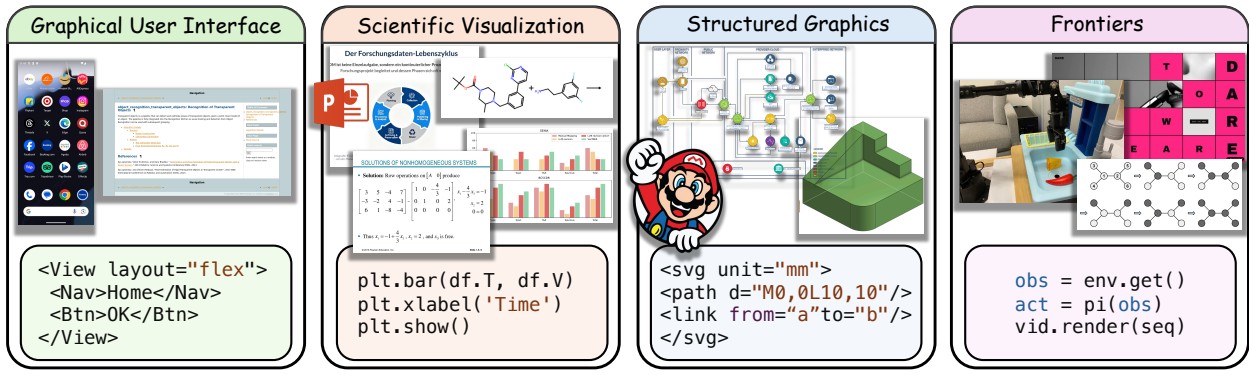

Figure 2: Overview of the Multimodal Code Intelligence landscape. The field is organized in this survey into four domains: (1) Graphical User Interface, transforming visual UI designs into frontend code (e.g., React/HTML); (2) Scientific Visualization, converting charts and scientific documents into plotting scripts (e.g., Matplotlib); (3) Structured Graphics, representing vector graphics and diagrams as structured code (e.g., SVG, CAD); and (4) Frontier Tasks and Frameworks, encompassing emerging applications such as vision-based programming, embodied control and video generation logic.

on recent work from 2022–2026 and earlier benchmark or dataset papers that define important tasks or evaluation protocols. Because the field evolves quickly, we maintain both the survey and the accompanying repository with newly released papers, benchmark links, and project resources.

In total, this survey covers a broad body of papers across four main domains. We include works in which visual inputs, visual outputs, or visually grounded states are used to generate, edit, verify, execute, or reason with code, as well as works in which code serves as a renderable visual representation, an intermediate reasoning trace, an executable artefact, or an action interface. We exclude purely language-driven code generation and software-engineering issue-resolution papers unless the task uses visual evidence or evaluates code through rendered, visually inspectable artifacts or visually grounded execution. Each work is coded by domain, task formulation, role of code, and evaluation signal. Ambiguous cases are resolved by checking the original task definition, benchmark protocol, or method objective. Figures 3 and 1 summarize the resulting domain coverage and quarterly publication trend. LLM tools were used only as assistive aids for drafting, consistency checking, and metadata organization; inclusion decisions, taxonomy assignments, and final descriptions were manually checked by the authors.

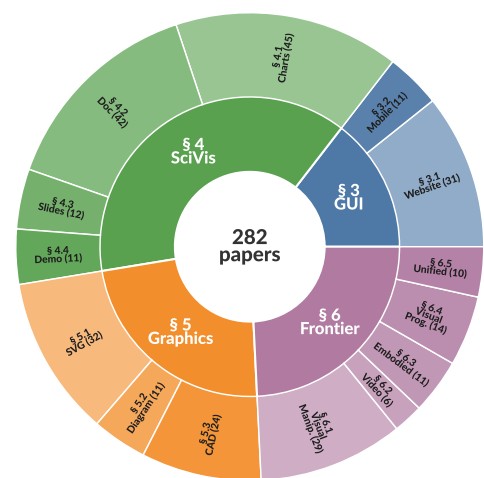

Figure 3: Survey coverage in Sections 3–6. The sunburst reports subdomain citation counts after de-duplication.

# 2 Task Formulation

In this section, we provide a formal taxonomy for Multimodal Code Intelligence. We define the core tasks by categorizing them into visual-to-code synthesis and code-centric reasoning paradigms.

## 2.1 NL2Code Preliminaries

To establish a formal baseline for multimodal extensions, the conventional NL2Code paradigm aims to synthesize an executable program $\mathcal{C}$ given a natural language description $\mathcal{T}$. Formally, this task is modeled

as a mapping function

$$\mathcal{C} = \mathrm{LLM}(\mathcal{T}). \tag{1}$$

While effective for logic-centric tasks, this unimodal formulation lacks the capacity to perceive spatial requirements, which are often essential in scenarios where intent is intrinsically tied to visual information.

## 2.2 Multimodal Code Synthesis

We define Multimodal Code Synthesis as the process of generating or modifying code where visual context $\mathcal{I}$, rendered feedback, or visually specified intent is central to the task. Depending on the initial state and the underlying manipulation intent, we delineate three sub-tasks:

**Multimodal Direct Generation.** In this paradigm, the model is provided with a visual context $\mathcal{I}$ (e.g., a chart, GUI screenshot, document page, design state, or rendered example) alongside a textual prompt $\mathcal{T}_{\mathrm{desc}}$. The objective of *Direct Generation* is to synthesize code that produces the requested visual artifact after execution, either by reconstructing a visible reference or by realizing a multimodal specification. The generation process is formulated as:

$$\mathcal{C}_{\mathrm{gen}} = \mathrm{MLLM}(\mathcal{I}, \mathcal{T}_{\mathrm{desc}}) \tag{2}$$

where $\mathcal{T}_{\mathrm{desc}}$ specifies the target artifact and the expected code form. In screenshot-to-code settings, this formulation resembles image-to-code reconstruction; in NL-to-chart, document, presentation, or demonstration settings, the visual artifact may be specified by text, context, or target rendering requirements. Its primary bottleneck is visual fidelity: the generated program must reproduce layout, geometry, style, and visible content after execution. However, visual fidelity is only the first layer of correctness. Later sections show that a visually similar program can still contain wrong chart data, non-editable SVG paths, invalid CAD constraints, broken UI handlers, or unsupported scientific semantics, which is why direct generation must eventually be paired with structure- and execution-aware validation.

**Instruction-driven Code Editing.** To further expand the task scope and leverage the instruction-following capabilities of MLLMs, recent works have explored multimodal code editing. This task requires the model to manipulate visual content based on specific user instructions. Current editing paradigms generally fall into two categories: (1) Text-guided editing, where modification intents are conveyed solely through natural language; and (2) Visual-prompt editing, where textual requirements are combined with visual prompts $\mathcal{V}$ (e.g., bounding boxes or encircling target regions) to precisely localize target elements. Formally, given an initial image or visual state $\mathcal{I}$, a textual editing instruction $\mathcal{T}_{\mathrm{edit}}$, an optional visual prompt $\mathcal{V}$ (where $\mathcal{V} = \emptyset$ in text-guided settings), and an optional source or intermediate representation $\mathcal{S}$, the model must generate target edited code $\mathcal{C}_{\mathrm{edited}}$ that renders the desired visual state. Source-agnostic variants infer the target program from pixels, while code-aware or tool-based variants edit existing source, templates, JSON states, slide objects, or design-tool representations. The editing process is formulated as:

$$\mathcal{C}_{\mathrm{edited}} = \mathrm{MLLM}(\mathcal{I}, \mathcal{T}_{\mathrm{edit}}, \mathcal{V}, \mathcal{S}) \tag{3}$$

This task evaluates the model's capacity for visual reasoning, precise spatial grounding, and counterfactual code synthesis with or without structural guidance.

**Reference-based Code Refinement.** While editing focuses on manipulating content based on external instructions, multimodal code refinement focuses on error correction and quality improvement. Drawing upon advancements in multi-turn debugging, this task provides the model with an explicit starting point: a potentially flawed code draft $\mathcal{C}_{\mathrm{draft}}$. The goal is to generate a refined version $\mathcal{C}_{\mathrm{refined}}$ that aligns the draft with the visual reference $\mathcal{I}$ or satisfies specific constraints $\mathcal{T}_{\mathrm{refine}}$. The formulation is:

$$\mathcal{C}_{\mathrm{refined}} = \mathrm{MLLM}(\mathcal{I}, \mathcal{T}_{\mathrm{refine}}, \mathcal{C}_{\mathrm{draft}}) \tag{4}$$

In contrast to the source-code-agnostic nature of editing, refinement allows the model to leverage $\mathcal{C}_{\mathrm{draft}}$ as a structural prior, focusing its computational capacity on precise alignment and functional debugging.

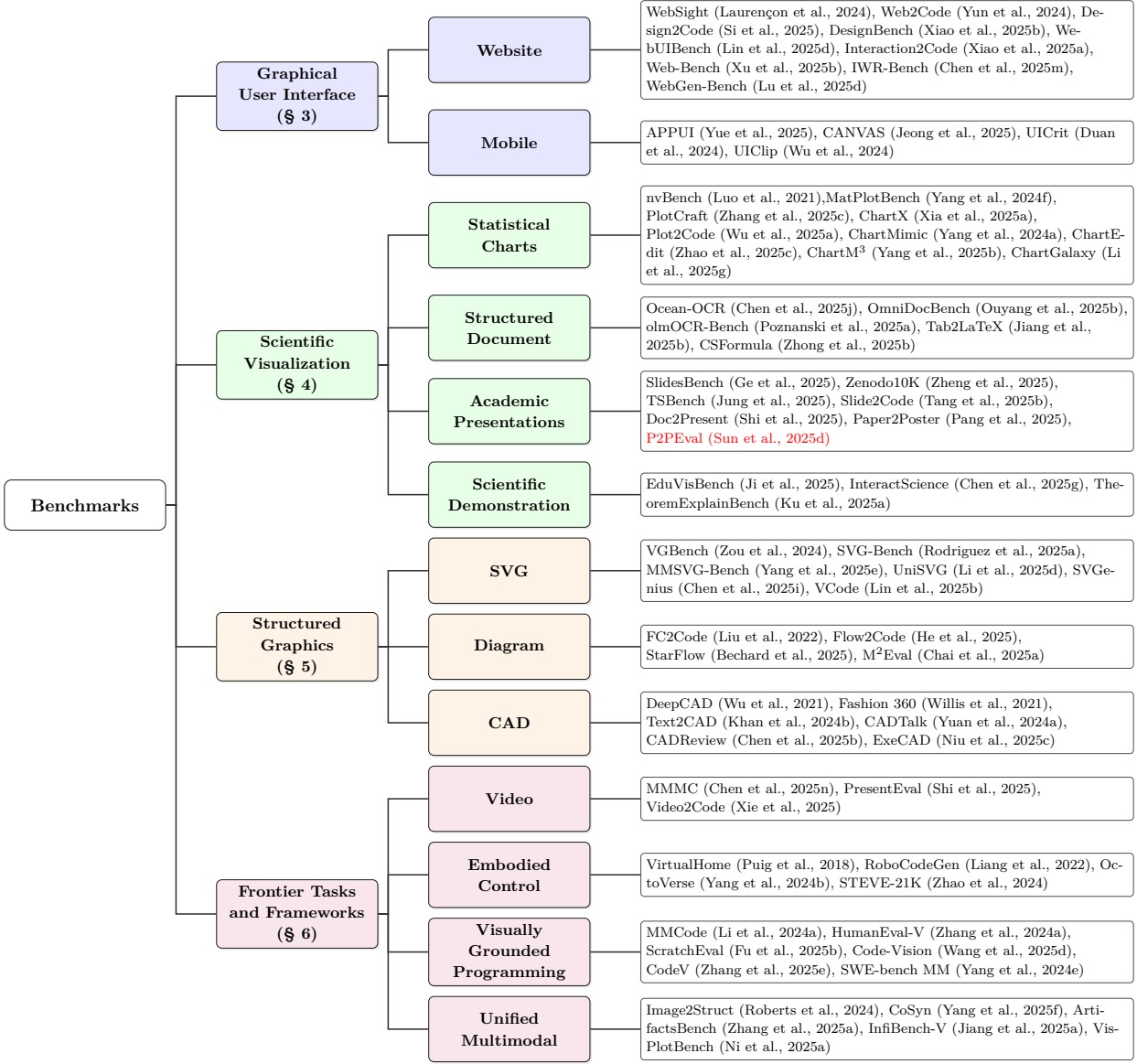

Figure 4: Taxonomy of representative benchmarks for multimodal code intelligence. We categorize datasets into four domains: Graphical User Interface (§ 3), Scientific Visualization (§ 4), Structured Graphics (§ 5), and Frontier Tasks and Frameworks (§ 6). Leaf nodes list selected benchmarks that cover major task types and evaluation signals, while Section 2 defines the corresponding code roles and validation gaps.

## 2.3 Code-Centric Reasoning and Acting

Transcending the scope of visual synthesis, recent research has increasingly exploited executable code as a robust substrate for advanced reasoning and agentic interaction. In this subsection, we formalize the domain of code-aided reasoning, where code functions not as a visual end-product, but as a symbolic intermediary that bridges visual perception with logical deduction and environmental control. Specifically, we delineate this domain into two primary paradigms: Programmatic Tool-Use for complex reasoning, and Executable Policy for autonomous agents.

**Programmatic Tool-Use.** In this paradigm, code functions as an intermediate reasoning trace. Rather than performing end-to-end neural prediction, the model acts as a neuro-symbolic controller that decomposes

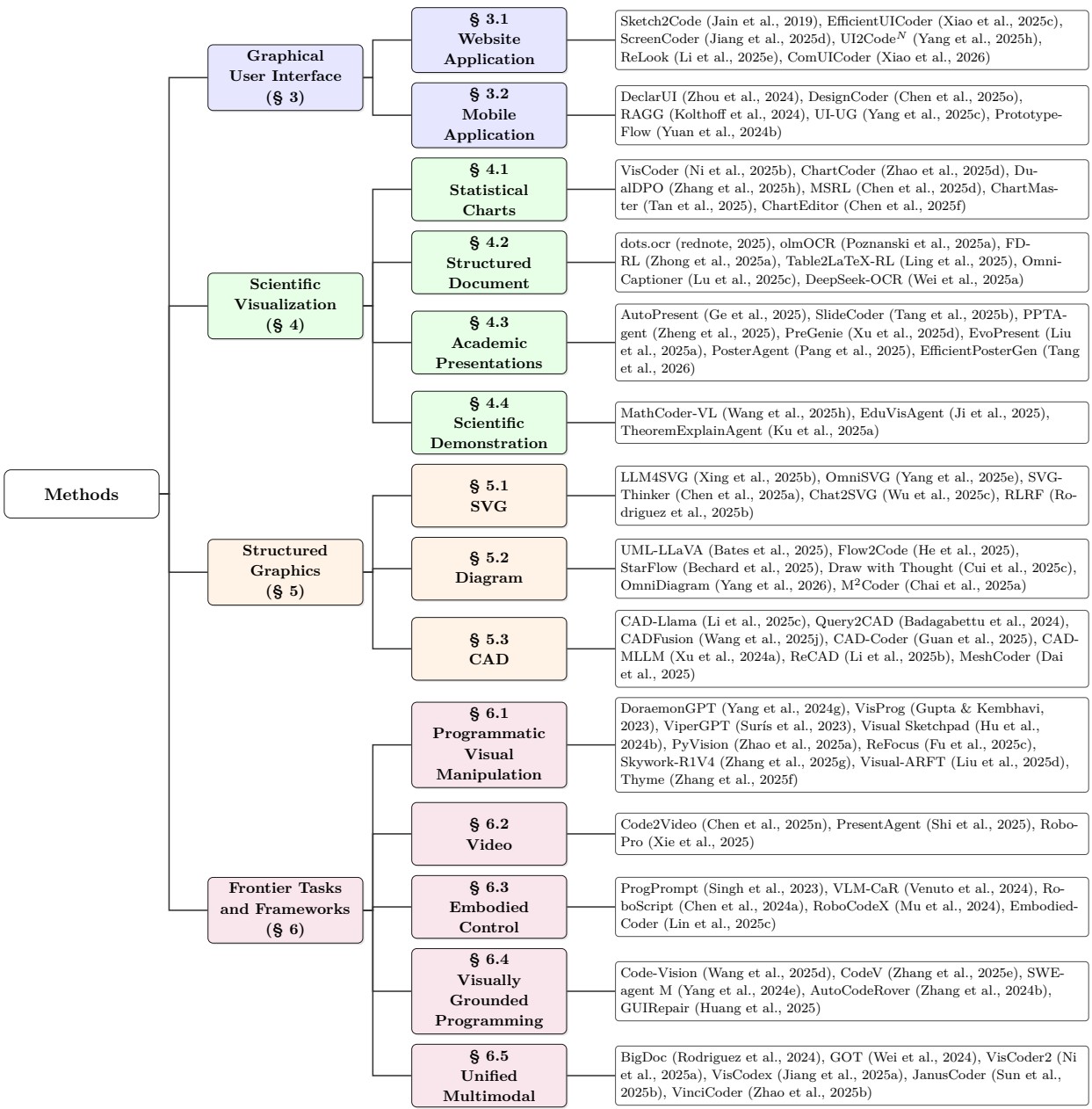

Figure 5: Taxonomy of representative multimodal code intelligence methods. The classification structure mirrors the benchmark taxonomy, spanning Graphical User Interface (§ 3), Scientific Visualization (§ 4), Structured Graphics (§ 5), and Frontier Tasks and Frameworks (§ 6). Leaf nodes list selected methods that illustrate major modeling routes, while Section 2 connects these routes to code roles and validation gaps.

a complex visual query $\mathcal{Q}$ into a modular program $\mathcal{C}_{\text{tool}}$. This program invokes external perceptual primitives (e.g., object detectors, OCR APIs) or performs precise symbolic calculations. Depending on the execution workflow, we categorize these methods into two distinct mechanisms:

- **Direct Programmatic Solving:** This approach adopts a deterministic framework where the task is resolved entirely via execution. The MLLM serves as a semantic parser to translate the natural language query into executable logic, and the execution result is treated as the final answer $\mathcal{A}$:

$$\mathcal{C}_{\text{tool}} = \text{MLLM}(\mathcal{I}, \mathcal{Q}), \quad \mathcal{A} = \text{Execute}(\mathcal{C}_{\text{tool}}, \mathcal{I}) \tag{5}$$

- **Tool-Augmented Visual Reasoning:** In this setting, the program acts as a perception enhancer to manipulate the visual input. The execution yields a processed view $\mathcal{I}' = \text{Execute}(\mathcal{C}_{\text{tool}}, \mathcal{I})$ (e.g., cropping or edge detection), which serves as an augmented context for subsequent inference:

$$\mathcal{A} = \text{MLLM}(\text{Execute}(\mathcal{C}_{\text{tool}}, \mathcal{I}), \mathcal{Q}) \tag{6}$$

By differentiating these pathways, the framework accommodates both rigorous symbolic derivation and flexible, perception-aware reasoning. We include such works when the generated code creates, inspects, transforms, or verifies visual evidence, or when the code trace itself is evaluated as part of the multimodal reasoning process.

**Executable Policy.** In the realm of sequential decision-making, code functions as a high-level policy $\pi$ that maps visual observations to structured actions. This paradigm applies to a wide spectrum of interactive environments, ranging from embodied robotics to digital GUI navigation. Unlike static text generation, the model synthesizes an executable action script $\mathcal{C}_{\text{policy}}$ based on the current state observation $\mathcal{O}_t$ and a high-level goal $\mathcal{G}$. This code-based policy utilizes control loops and API calls to enable temporally extended behaviors. The process is formulated as:

$$\mathcal{O}_{t+1} = \text{Env}(\pi(\mathcal{O}_t, \mathcal{G}), \mathcal{O}_t) \tag{7}$$

where the code-based policy $\pi(\mathcal{O}_t, \mathcal{G})$ interacts with the environment to drive the transition to the next state $\mathcal{O}_{t+1}$. In this context, code empowers the agent with a structured and generalizable action space, prioritizing functional correctness and goal attainment over mere visual fidelity.

## 3 Graphical User Interface

Graphical User Interface code generation translates visual designs into executable implementations, spanning both web and mobile platforms. Under the Section 2 taxonomy, GUI tasks primarily instantiate direct generation, editing, and refinement, while interactive web and mobile benchmarks connect these formulations to action replay and environment-transition checks. For UI-to-code, the evaluation environment should connect generated code, rendered visual states, user actions, and measurable correctness signals. We first review website application benchmarks and methods, then examine the distinct challenges of mobile application generation.

### 3.1 Website Application

Web-to-code provides the clearest GUI setting because HTML, CSS, JavaScript, browsers, and WebDriver form a shared loop in which webpages can be generated, rendered, inspected, and interacted with at scale. Bridging the semantic gap between high-level visual designs and their programmatic implementations, website-to-code (Web-to-Code) generation automates the translation of visual interfaces into standardized web technologies. In this setting, VLMs utilize visual reasoning to extract structural and stylistic attributes from raw pixel inputs, thereby achieving faithful reconstruction (Luera et al., 2024).

This execution loop makes evaluation practical, but it can also overemphasize rendered similarity when functional correctness is not tested. We therefore review benchmarks by the correctness signal exposed through the browser, and then trace methods from imitation-based reconstruction to agentic decomposition and feedback-driven executable verification. The standard web-to-code workflow is illustrated in Figure 6, with key benchmarks summarized in Table 1.

### 3.1.1 Website Code Generation Benchmarks

From this perspective, the browser is not only a renderer but also an evaluator, and benchmarks differ in which part of the generated system they observe through it. We group web benchmarks by three dominant correctness signals: static rendering, executable interaction, and specialized preference or agent-task evaluation.

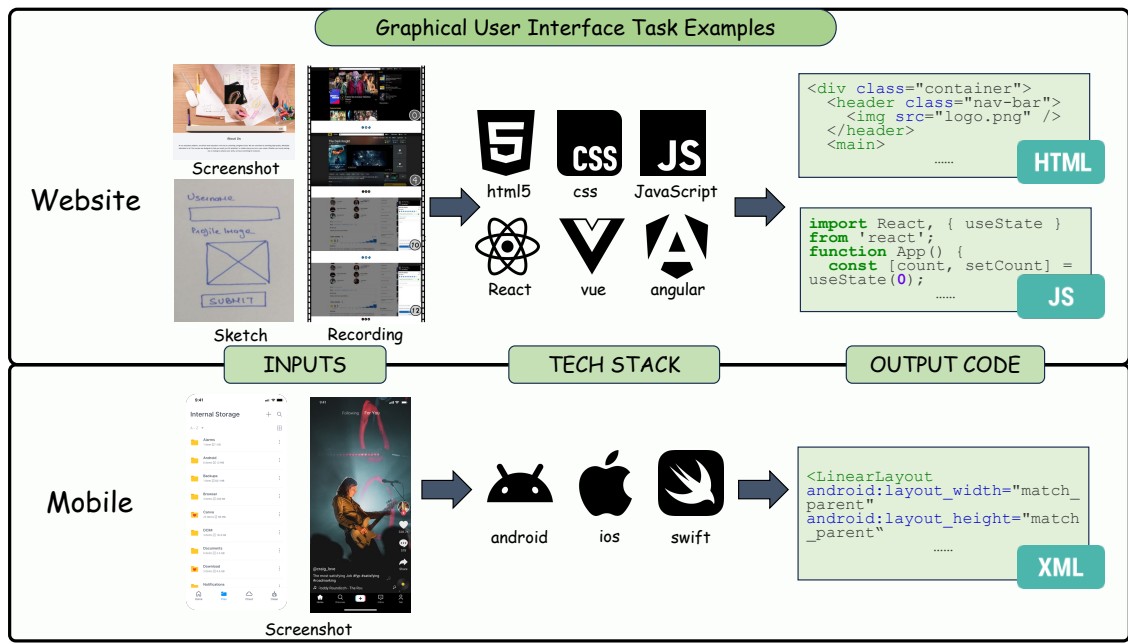

Figure 6: Examples of GUI code generation tasks for website and mobile applications.

Static benchmarks primarily evaluate structural and visual alignment between generated webpages and ground-truth designs. They utilize browser rendering as the main evidence, which makes evaluation scalable but biases the field toward appearance-level correctness. Large-scale reconstruction benchmarks such as WebSight (Laurençon et al., 2024) and Web2Code (Yun et al., 2024) build synthetic screenshot-code pairs, while Design2Code (Si et al., 2025) and WebCode2M (Gui et al., 2025) move the setting toward real websites and Common Crawl data. More diagnostic benchmarks inspect different parts of the rendered page: Vision2UI (Gui et al., 2024) targets DOM-tree recovery, IW-Bench (Guo et al., 2025a) measures element-level layout accuracy, and WebRenderBench (Lai et al., 2025) and WebGen-V Bench (Wang et al., 2025i) emphasize rendered fidelity. DesignBench (Xiao et al., 2025b), WebUIBench (Lin et al., 2025d), and Full-Front (Sun et al., 2025a) further broaden the signal toward editing, repair, aesthetic quality, and perceptual comprehension.

Complementing static assessments, dynamic evaluation protocols verify whether the generated code implements the intended website functionality and interactivity as visual-similarity benchmarks become easier to saturate and harder to audit for potential data leakage. In these benchmarks, the browser further serves as an interaction executor. Interaction2Code (Xiao et al., 2025a) evaluates reactive interface prototyping, MR-Web (Wan et al., 2024) adds multi-page navigation and backend routing, and Web-Bench (Xu et al., 2025b) tests sequential development tasks. IWR-Bench (Chen et al., 2025m) targets interactive reconstruction from video inputs, whereas WebGen-Bench (Lu et al., 2025d) uses a GUI agent to simulate user interactions and check functionality through execution-based cases. Specialized arenas provide adjacent rather than directly comparable evidence. DesignArena (The Intelligence Company, 2025) emphasizes human preference over generated UI and design artifacts, while the Code Arena WebDev leaderboard (Arena AI, 2026) ranks models on front-end web development tasks, including agentic coding workflows that require multi-step reasoning and tool use. These settings should be reported separately from web-to-code reconstruction benchmarks because preference, web-development task performance, and leakage controls expose different evidence.

### 3.1.2 Website Code Generation Methods

The methodological landscape of Web-to-Code has evolved from monolithic generation to sophisticated, interactive systems. The shared browser environment explains this progression: imitation-based models learn from rendered pairs, agent workflows decompose interface construction into inspectable steps, and feedback-based methods utilize rendering and interaction feedback as training or repair signals.

Early approaches, such as Sketch2Code (Jain et al., 2019), relied on hand-crafted object detection models and UI parsers to translate sketches into intermediate representations. With the advent of VLMs, a major line of work adopts direct SFT rather than a single field-wide pivot. Pioneering efforts, such as WebSight (Laurençon et al., 2024) and Web2Code (Yun et al., 2024), establish foundational baselines by training on large-scale synthetic datasets. Design2Code (Si et al., 2025) and WebCode2M (Gui et al., 2025) further scale this approach by incorporating diverse real-world web data to enhance model robustness. To improve the computational efficiency of these SFT-based models, EfficientUICoder (Xiao et al., 2025c) introduces a bidirectional token compression framework, significantly reducing inference overhead. However, despite these advances, single-model architectures often struggle to generalize in complex, open-ended scenarios.

Another line develops collaborative agent-based frameworks that decompose generation into inspectable subproblems instead of relying on a single open-loop model. Such multi-agent architectures factor UI code generation into visual grounding, layout planning, implementation, review, and execution-based refinement. ScreenCoder (Jiang et al., 2025d) proposes a modular architecture comprising grounding, planning, and generation units to enhance task interpretability. Frontend Diffusion (Ding et al., 2025) decouples the workflow into distinct design, coding, and review stages, thereby improving system modularity. TDDev (Wan et al., 2025) extends this philosophy to full-stack development, introducing a multi-agent system based on Test-Driven Development (TDD) that orchestrates requirement extraction, test generation, and iterative refinement. Supporting these generation frameworks, Instruct4Edit (Dang et al., 2025) employs LLMs to programmatically synthesize high-quality datasets specifically tailored for code editing tasks, while ComUICoder (Xiao et al., 2026) applies semantic-aware segmentation for component-based UI code generation, improving code reusability and maintainability.

Recognizing the constraints of single-turn, open-loop generation, contemporary works increasingly integrate iterative visual feedback, critic-based repair, agent replay, and a smaller subset of RL-style optimization to improve functional and visual alignment. This progression reflects an escalating demand for correctness signals beyond text loss, but browser feedback remains partial unless it exercises behavior across states. WebGen-Agent (Lu et al., 2025d) pioneers the incorporation of multi-level visual feedback, establishing a closed-loop cycle of code generation, execution, and optimization. UI2CodeN (Yang et al., 2025h) unifies generation, editing, and polishing capabilities. In the realm of feedback-driven optimization, ReLook (Li et al., 2025e) introduces a visual-driven framework, employing a VLM as a critic to orchestrate a diagnosis-and-optimization loop. Coder-CUA (Lin et al., 2025a) advocates for a shift from human-centric to agent-centric evaluation, delegating assessment to a Computer-Use Agent that verifies functional correctness through task execution rather than static visual similarity. By utilizing a code agent to initialize and refine UIs and a Computer-Use Agent (CUA) to assess them via navigation success and task solvability, this method redefines the interface development workflow. Such feedback-driven methods are most reliable when visual rewards, task-completion rewards, and code-level checks are reported separately, since visual critics can still overfit static appearance while CUA scores depend on task coverage, environment stability, and leakage controls.

**Scope and Trajectory.** The central bottleneck in web-to-code research is that the browser can support execution-based verification, but most evaluations still observe only its rendered surface. Early progress therefore concentrated on screenshot-to-HTML reconstruction because rendered similarity is scalable and easy to standardize. This signal is useful for layout, typography, and style, but it is not a reliable proxy for application correctness because a web UI is correct over a set of states, not only in a single viewport. A page can match the reference image while omitting event handlers, losing state transitions, breaking routing, or encoding the interface in brittle code that cannot support revision and reuse.

Recent benchmarks consequently shift from static page generation toward executable interfaces, where the failure modes become more explicit. In the reported IWR-Bench setting, models can obtain a visual fidelity score of 64.25% while reaching only 24.39% on interactive function (Chen et al., 2025m). This gap indicates that rendered similarity and interaction correctness measure different capabilities. Static screenshots hide event logic, latent state changes, component boundaries, data flow, and error handling. As a result, optimizing only for visual overlap can reward implementations that look correct in a fixed viewport but fail under user actions or code-level inspection.

| Benchmark | Year | Data Source | Test Instances | Evaluation Metrics |
|---|---|---|---|---|
| *Static Benchmark* | | | | |
| WebSight (Laurençon et al., 2024) | 2024 | Synthetic | 823k | BLEU, TreeBLEU, SSIM |
| Web2Code (Yun et al., 2024) | 2024 | Synthetic+Real | 884.7k | Visual similarity, Code match |
| Design2Code (Si et al., 2025) | 2024 | Real | 484 | CLIP, Block match, Visual sim |
| WebCode2M (Gui et al., 2025) | 2024 | Real | 20k | TreeBLEU, SSIM, Error rate |
| Vision2UI (Gui et al., 2024) | 2024 | Real | 20k | TreeBLEU, SSIM |
| IW-Bench (Guo et al., 2025a) | 2024 | Real+Synthetic | 1.2k | Element/Layout Accuracy |
| FullFront (Sun et al., 2025a) | 2025 | Real+Synthetic | 400 | Gemini Visual Score, Code Score |
| WebRenderBench (Lai et al., 2025) | 2024 | Real | 45.1k | RDA, GDA, SDA |
| WebGen-V Bench (Wang et al., 2025i) | 2024 | Real | 647 | Layout/Style consistency |
| DesignBench (Xiao et al., 2025b) | 2025 | Real | 900 | Generation/Editing/Repair |
| *Dynamic Benchmark* | | | | |
| Interaction2Code (Xiao et al., 2025a) | 2024 | Real | 97 | Interaction success rate |
| MRWeb (Wan et al., 2024) | 2024 | Real | 500 | Functionality completeness |
| Web-Bench (Xu et al., 2025b) | 2025 | Real | 50 projects | Pass@k |
| WebGen-Bench (Lu et al., 2025d) | 2024 | Real | 101 | Accuracy, Appearance score |
| IWR-Bench (Chen et al., 2025m) | 2025 | Real | 113 | Interaction fidelity |
| *Specialized Benchmark* | | | | |
| UIClip (Wu et al., 2024) | 2024 | Synthetic+Real | 2.3M+1.2k | Design quality score |
| AUI-Gym (Lin et al., 2025a) | 2024 | Synthetic | 52 apps | Function Completeness, CUA SR |
| WebVIA (Xu et al., 2025c) | 2025 | Synthetic+Real | 11k | Task completion rate |

Table 1: Representative benchmarks in the UI-to-Code domain. Static benchmarks focus on visual similarity between generated and reference UIs, while dynamic benchmarks evaluate functional correctness through automated interaction testing.

The trajectory of the field should therefore treat the browser as a controlled verification environment rather than only a rendering engine. Stronger protocols need to pair visual targets with structured requirements, replayable user actions, state assertions, and code-level checks. This design separates what works from what fails: rendered metrics capture perceptual alignment, interaction tests capture behavioral correctness, and code analysis captures maintainability and architectural validity. Metric reliability depends on reporting these signals separately and on testing whether visual gains transfer to interaction success and robust component structure.

### 3.2 Mobile Application

In contrast to the web setting, where code, rendering, interaction, and evaluation can be connected by browsers and WebDriver, Mobile-to-Code generation has a more fragmented execution environment. Native applications are compiled binaries with no publicly crawlable source and no single rendering or interaction environment shared across platforms. This fragmentation is a structural practical constraint in current mobile evaluation, and it shapes how benchmarks and methods choose proxy signals.

#### 3.2.1 Mobile Code Generation Benchmarks

Existing mobile benchmarks have not yet established the full code-render-interaction loop used in web evaluation. Native app generation depends on platform-specific build systems, device or emulator execution, and interaction instrumentation, which makes end-to-end evaluation difficult to standardize. Current benchmarks therefore rely on proxy settings that expose only partial evidence of correctness.

Within this proxy-based setting, the first strategy is artifact-centric evaluation, where mockups or design-tool states stand in for native applications. APPUI (Yue et al., 2025) establishes a comprehensive benchmark comprising 1.1k pairs of rendered images and code across 12 application categories, specifically curated for the reconstruction of static single-page applications from design mockups. APPUI therefore utilizes paired mockups as a substitute for inaccessible native source. Extending the scope to tool-assisted design workflows, CANVAS (Jeong et al., 2025) targets Figma-based mobile UI design with 598 tool-driven tasks sampled from

3.3k human-crafted designs. Together, APPUI and CANVAS cover artifact reconstruction and design-tool editability, leaving native runtime behavior outside the benchmark scope.

The second strategy is evaluator-centric evaluation, where critiques or learned scoring models replace executable apps as the source of quality signals. UICrit (Duan et al., 2024) introduces a dataset of 3k critiques annotated with bounding boxes and design-quality ratings, serving as a critical resource for alignment via reward modeling. In parallel, UIClip (Wu et al., 2024) provides a screenshot-based scoring model. By utilizing contrastive learning, it functions as an effective reranker for mobile synthesis, surpassing simple pixel-level matching metrics. These evaluator-based benchmarks make comparison scalable, but their scores reflect preference or surface quality rather than executable correctness. Thus, the mobile benchmark comparison should be read through a fixed template: APPUI observes static mockup reconstruction, CANVAS observes design-tool editability, UICrit observes localized critique quality, and UIClip observes screenshot-level preference, while none directly verifies native runtime behavior.

### 3.2.2 Mobile Code Generation Methods

Given the proxy settings above, mobile code generation methods mainly try to make partial correctness signals observable rather than close the native runtime loop directly. Unlike web-to-code, which can utilize millions of crawlable HTML pages and browser-based feedback, mobile-to-code must extract maximal signal from screenshots, UI hierarchies, design states, and small interaction traces. Structure-aware and retrieval-augmented methods address this problem by making layout organization or corpus precedent explicit. DeclarUI (Zhou et al., 2024) combines component segmentation with a Page Transition Graph to capture multi-screen navigation logic, further utilizing iterative compilation checks to rectify syntax errors. DesignCoder (Chen et al., 2025o) explicitly models UI hierarchy via a UI Grouping Chain and introduces a vision-aware autonomous repair mechanism to refine code post-rendering. RAGG (Kolthoff et al., 2024) retrieves relevant references from the Rico dataset (Deka et al., 2017) and employs self-critique loops to synthesize more structurally grounded UIs. These signals improve structural recovery and pattern consistency, but they still leave gesture semantics, platform widgets, state-dependent behavior, novel requirements, and maintainability only partially verified.

Other methods construct feedback or editable prototype spaces as more operational proxies. UI-UG (Yang et al., 2025c) incorporates RL to jointly optimize UI understanding and generation quality within a single model, showing how learned rewards make optimization tractable while risking overfitting to measurable layout or quality signals that miss user intent and native runtime behavior. PrototypeFlow (Yuan et al., 2024b) focuses on creating high-fidelity prototypes (e.g., SVG/JSON) rather than final application code, providing editable intermediate checkpoints to facilitate a flexible mobile-oriented creation process. Generative Interface (Chen et al., 2025c) targets NL-to-UI generation as a mobile-oriented proxy setting. Unlike standard native Mobile-to-Code tasks, it utilizes an LLM to generate task-specific interactive UIs (e.g., HTML/JavaScript) from user queries through structured representations and iterative refinement, prioritizing user experience over pixel-perfect reconstruction. Together, these methods make optimization, inspection, and handoff easier, but they remain prototype-level substitutes unless connected to native runtime constraints or interaction tests.

**Scope and Trajectory.** Taken together, mobile code generation currently advances through proxies rather than a closed native execution loop. Benchmarks approximate correctness with artifacts, design-tool states, critiques, or learned scores, while methods make partial correctness observable through structure, retrieval, rewards, and editable intermediate representations.

These signals expose visual reconstruction, editability, preference, component structure, and constrained feedback, but they do not by themselves verify native compilation, platform widgets, gesture handling, state changes, accessibility constraints, or maintainable implementation. The trajectory of the field should therefore be to make each proxy signal explicit and progressively connect benchmark signals and method feedback to native runtime constraints, interaction lifecycles, and implementation validity.

> **Takeaway.** Across web and mobile UI generation, evaluation must bridge static visual reconstruction and dynamic runtime behavior. Visual similarity is useful for layout and style. Still, it is not enough to show that an interface works across user actions, state changes, responsive views, accessibility constraints, or component reuse. Web tasks have the clearest execution substrate via browsers, WebDriver, DOM inspection, and runtime debugging, whereas mobile tasks rely more heavily on proxy signals from artefacts, design tools, UI hierarchies, retrieval, rewards, and editable intermediates. These signals should be reported as complementary evidence rather than complete validation. This domain therefore motivates multi-state verification, where validators connect screenshots, generated code, executed UI states, user actions, and implementation structure.

## 4 Scientific Visualization

We broadly to cover scientific visual-code artifacts rather than only traditional plotting. These tasks require generated code to preserve the scientific semantics behind a visual artifact, not only its rendered appearance. In the Section 2 taxonomy, most tasks are direct generation or refinement, while scientific demonstrations also approach programmatic tool-use because generated code can act as an explanatory or validation trace. This section reviews statistical charts, structured documents, academic presentations, and scientific demonstrations, where code serves as a renderable and inspectable representation for scientific meaning. Across these settings, correctness depends on whether the generated artifact preserves data, structure, argument flow, equations, and domain constraints in forms that can be executed, edited, or validated.

### 4.1 Statistical Charts

Chart generation is structured around two task formulations that impose different constraints on model capabilities. NL-to-Chart generation maps ambiguous natural language intent to executable visualization code, while Chart-to-Code generation recovers the data, visual encoding, and plotting logic implied by a rendered chart. These tasks are not symmetric: the former is an under-specified synthesis problem where multiple visualizations can satisfy one query, whereas the latter is a constrained reconstruction problem where many code programs may render the same chart but must preserve the same data semantics and visual encodings. This asymmetry explains why methods that excel at one task often fail at the other, and why the field has developed different methodological trajectories for each direction. The general task formulation for scientific visualization is depicted in Figure 7.

#### 4.1.1 Chart Code Generation Benchmarks

Chart benchmarks follow the two task formulations introduced above. NL-to-Chart benchmarks start from textual analytic intent, so they evaluate whether the generated code executes and whether the rendered chart is semantically appropriate for the query. Chart-to-Code benchmarks start from a rendered chart, so they evaluate whether the generated program reconstructs the visual encodings, recovered data, and editable or executable plotting logic.

Benchmarks in the NL-to-Chart domain primarily focus on the fidelity and executability of code synthesized from textual instructions. Their core difficulty is not only whether code runs, but whether the rendered chart satisfies an underspecified analytic intent. Pioneering efforts, such as nvBench (Luo et al., 2021) and VisEval (Chen et al., 2024b) use synthetic datasets to evaluate code executability and output validity. In contrast, MatPlotBench (Yang et al., 2024f) and PandasPlotBench (Galimzyanov et al., 2025) curate evaluation samples from real-world galleries and use an LLM-as-a-judge mechanism for semantic assessment. More recently, the field has shifted toward greater complexity (Lu et al., 2025a; Rahman et al., 2025). For example, nvBench 2.0 (Luo et al., 2025) introduces a large-scale dataset characterized by one-to-many mappings and complex reasoning traces. Simultaneously, PlotCraft (Zhang et al., 2025c) proposes a benchmark with 982 instances that incorporates multi-turn refinement tasks, moving beyond standard single-turn generation.

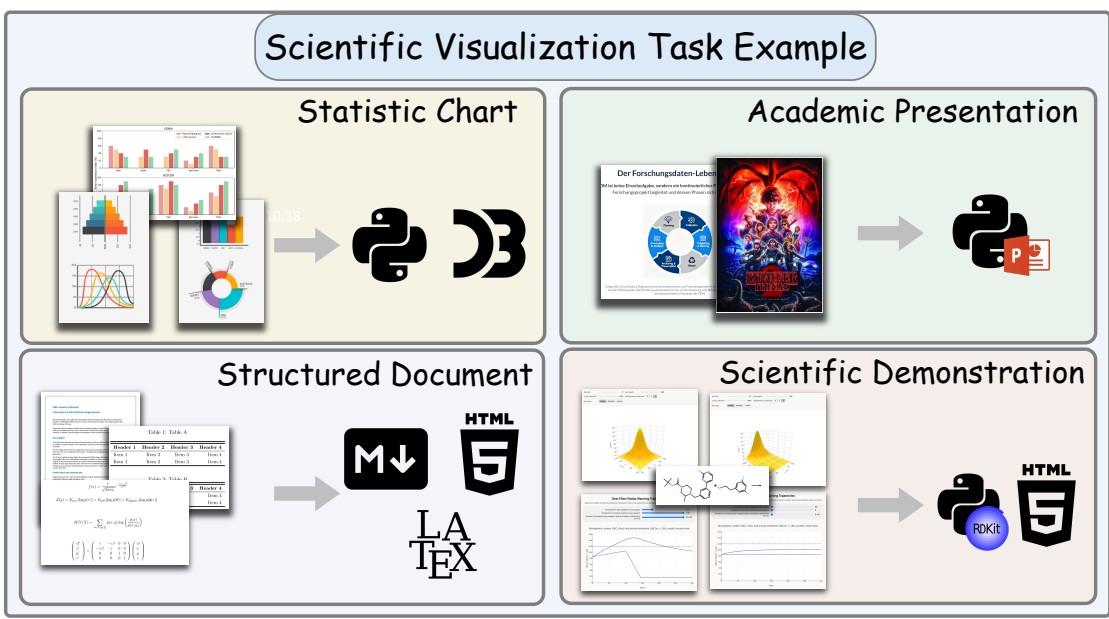

Figure 7: Examples of scientific visualization code generation tasks, including charts, documents, presentations, and demonstrations.

In parallel, the Chart-to-Code setting aims to reverse-engineer executable code directly from visual chart inputs. Here, the analytical target is reconstruction fidelity rather than intent satisfaction: the model must recover the data, visual encoding, and executable plotting logic implied by the input chart. Existing reconstruction benchmarks vary significantly in data composition, ranging from synthetic to real-world charts. ChartX (Xia et al., 2025a) uses synthetic chart images and assesses generation quality via an LLM-based evaluation framework, while Plot2Code (Wu et al., 2025a) incorporates 132 real-world images and evaluates performance using text-matching metrics against reference code. A separate but related editing setting evaluates whether models can modify an existing chart according to textual or visual instructions while preserving unaffected data and encodings. ChartMimic (Yang et al., 2024a) introduces data-driven editing tasks alongside direct generation, supported by manually annotated reference code for rigorous rule-based evaluation. ChartEdit (Zhao et al., 2025c) enriches the editing landscape by introducing diverse instruction types, supported by 1.4k human-annotated instructions on 233 real-world charts. ChartM$^3$ (Yang et al., 2025b) and ChartEditVista (Chen et al., 2025f) further develop multimodal chart editing by integrating textual instructions and visual indicator-guided mechanisms. Furthermore, ChartGalaxy (Li et al., 2025g) extends the target domain to infographic charts focusing on D3.js. Chart2Code (Tang et al., 2025a) introduces a hierarchical task structure ranging from direct generation to multi-faceted editing. Representative Chart-to-Code and NL-to-Chart benchmarks are summarized in Table 2.

### 4.1.2 Chart Code Generation Methods

Methodological development mirrors the benchmark split. NL-to-Chart methods target intent alignment through planning, visual feedback, and instruction tuning, while Chart-to-Code methods target reconstruction fidelity through chart-code data, rendering-aware feedback, and editing-oriented optimization.

For NL-to-Chart, methods treat text prompts as underspecified analytic intents and use feedback or training to refine code toward an acceptable visualization. MatPlotAgent (Yang et al., 2024f) pioneers the use of visual feedback to iteratively refine the generated Matplotlib code. VisPath (Seo et al., 2025) proposes a multi-path reasoning and feedback mechanism, while nvAgent (Ouyang et al., 2025a) and AMACE (Namgoong et al., 2025) introduce collaborative multi-agent workflows to tackle complex visualization tasks. Similarly, Doc2Chart (Jain et al., 2025) extends this to document-to-chart scenarios via an interactive protocol.

Training-oriented methods embed visualization expertise directly into model parameters, but their signal still has to approximate whether the executed chart satisfies the requested analysis. Text2Chart31 (Zadeh et al.,

| Benchmarks | Year | Test Instances | Key Features | Correctness Signal |
|---|---|---|---|---|
| *Chart-to-Code Benchmark* | | | | |
| Plot2Code (Wu et al., 2025a) | 2025 | 132 | Real-world Images | Reference rendering/code |
| ChartMimic (Yang et al., 2024a) | 2024 | 4.8k | Code/Visual Evaluation | Data, layout, color |
| ChartEdit (Zhao et al., 2025c) | 2025 | 1.4k | Diverse Editing Types | Instruction edit success |
| ChartM$^3$ (Yang et al., 2025b) | 2025 | 1k | Visually-guided Editing | Visual-change fidelity |
| *NL-to-Chart Benchmark* | | | | |
| MatPlotBench  (Yang et al., 2024f) | 2024 | 100 | Human Verification | Semantic chart quality |
| VisEval (Chen et al., 2024b) | 2024 | 2.3k | Large-Scale Coverage | Executability and validity |
| Text2Vis (Rahman et al., 2025) | 2025 | 2k | Diverse Data Science Queries | Intent-query alignment |
| PlotCraft(Zhang et al., 2025c) | 2025 | 982 | Multi-turn Generation | Task compliance and quality |

Table 2: Representative benchmarks in the statistical chart generation domain. We categorize these benchmarks into Chart-to-Code and NL-to-Chart settings. The correctness-signal column shows that Chart-to-Code benchmarks tend to observe reconstruction fidelity, while NL-to-Chart benchmarks must approximate intent satisfaction through execution, semantic judgment, and multi-turn task compliance.

2024) uses Proximal Policy Optimization (PPO) (Schulman et al., 2017) combined with automated feedback and cycle consistency to optimize visualization instruction-code pairs. VisCoder (Ni et al., 2025b) introduces VisCode-200K, a dataset tailored for Python-based visualization and self-correction. More recently, Step-Text2Vis (Luo et al., 2025) uses Step-DPO (Lai et al., 2024) on step-wise preference datasets to improve the logical granularity of the reasoning process. Similarly, PlotCraftor (Zhang et al., 2025c) synthesizes SynthVis-30K, a dataset integrating both single- and multi-turn samples, and reports substantial gains under its benchmark metrics when using the Qwen3-Coder-30B-A3B (Yang et al., 2025a) backbone for code synthesis.

For Chart-to-Code, methods first scale reconstruction through chart-code pairs and then add feedback to reduce the mismatch between code-token learning and rendered-chart correctness. Early efforts like MatCha (Liu et al., 2023a) focus on enhancing model capabilities through specialized pre-training strategies. Building on this, ChartLlama (Han et al., 2023) and ChartVLM (Xia et al., 2025a) establish robust baselines by fine-tuning MLLMs on synthetic data derived from closed-source LLMs. ChartMOE (Xu et al., 2024b) uses large-scale Chart-to-Code data to bridge the modality gap. To bolster Chart-to-Code generation specifically, ChartCoder (Zhao et al., 2025d) adopts a code-centric backbone (e.g., DeepSeek-Coder) and introduces a Snippet-of-Thought (SoT) reasoning strategy on its Chart2Code-160k dataset. Chart2Code53 (Niu et al., 2025d) further scales this by covering a diverse range of plotting functions and chart types. Beyond SFT, multi-agent approaches further improve code redrawing (Xu et al., 2025a; Jiang et al., 2025c), and METAL (Li et al., 2025a) uses multi-agent collaboration and test-time scaling to optimize redrawing accuracy. These methods improve executable reconstruction, but synthetic chart-code pairs can still miss the stylistic diversity and domain-specific conventions of professional charts.

Preference-driven methods address the remaining signal mismatch by optimizing programs against executed visual and data outcomes rather than textual code similarity alone. DualDPO (Zhang et al., 2025h) proposes a dual preference-guided refinement framework to synthesize training preference data, subsequently using DPO (Rafailov et al., 2023) for model optimization. MSRL (Chen et al., 2025d) and ChartMaster (Tan et al., 2025) adopt the GRPO (Shao et al., 2024) algorithm augmented with multimodal reward feedback, targeting both code granularity and visual structural alignment. However, they diverge in their data synthesis strategies: MSRL uses Gemini-2.0-Flash (Team et al., 2024) for text-driven plotting code generation, whereas ChartMaster employs an image-based method using Qwen2.5-VL-72B (Bai et al., 2025) to transform visual charts directly into code. Beyond direct generation, ChartReformer (Yan et al., 2024) pioneers natural language-driven editing by manipulating JSON structures, while ChartEditor (Chen et al., 2025f) introduces a rendering-aware reward signal using the Matplotlib backend for fine-grained supervision. The remaining caveat is reward reliability: rewards based mainly on visual similarity can prefer plausible renderings that contain wrong recovered data, incorrect grouping, misleading axes, or non-executable chart code, so robust optimization needs data-aware checks and execution-based validation alongside rendering feedback.

Adjacent chart reasoning work uses code as an intermediate representation rather than as the final generated artifact. ReachQA (He et al., 2024), ECD (Yang et al., 2025g), Chart-R1 (Chen et al., 2025e), and ChartReasoner (Jia et al., 2025) use code to render charts, structure reasoning, or expose intermediate checks. This makes code a verification substrate for inspecting chart evidence, connecting these works to Section 2's programmatic tool-use formulation.

**Scope and Trajectory.** Taken together, chart code generation is difficult because correctness is distributed across data fidelity, visual encoding fidelity, and intent fidelity. Charts are not ordinary images because their visual form compresses data operations and analytic claims into marks, scales, axes, legends, and layout choices. A generated chart can therefore look plausible while using wrong values, aggregations, encodings, or comparisons, and code similarity alone cannot show whether these semantics are preserved.

Current benchmarks remain too limited for real-world visualization scenarios that involve professional styling, interactive views, and domain conventions. The trajectory of the field should therefore move from single-signal evaluation toward multi-signal verification. Chart-to-Code needs visual reconstruction to be paired with data recovery and executable rerendering, while NL-to-Chart needs intent satisfaction and analytic appropriateness rather than proximity to one reference chart. Future benchmarks and training loops should report visual reconstruction, data accuracy, executable correctness, and design quality as separate signals, especially when operations, state changes, and data bindings provide additional evidence.

## 4.2 Structured Document

Structured documents serve as essential carriers of knowledge representation, encapsulating information through the intricate interplay of natural language, tabular data, and mathematical expressions. The task bottleneck is multi-grammar recovery, where page layout and reading order, text semantics, table/form grids, formula trees, and cross-region references must be preserved while being serialized into one output language. Unlike traditional Optical Character Recognition (OCR), which primarily focuses on extracting literal text, structured document code synthesis prioritizes reconstructing underlying logical architectures, such as hierarchical schemas, complex tables and forms, and nested formulas. To this end, research in this domain focuses on translating visual inputs into structured code representations, including Markdown, HTML, and LaTeX, thereby enabling precise, automated document parsing and semantic recovery.

### 4.2.1 Structured Document Code Generation Benchmarks

Benchmark design follows the multi-grammar bottleneck above and is organized into three canonical task formulations. Document-to-Markdown benchmarks such as OmniDocBench (Ouyang et al., 2025b), olmOCR-Bench (Poznanski et al., 2025a), and READoc (Li et al., 2025h) test full-page reading order, layout structure, and heterogeneous element recovery. Table-to-Code benchmarks such as PubTabNet (Zhong et al., 2020), FinTabNet (Zheng et al., 2021), and Table2LaTeX-RL (Ling et al., 2025) test grid structure, cell spans, and renderable markup, while Formula-to-LaTeX benchmarks such as IM2LaTeX-100K (Deng et al., 2017), UniMER-Test (Wang et al., 2024a), and CSFormula (Zhong et al., 2025b) test grammar-constrained sequence generation and rendered mathematical equivalence. Recent OCR-oriented benchmarks, including OCRBench v2 (Fu et al., 2025a) and KITAB-Bench (Heakl et al., 2025), further stress localization, multilingual document understanding, and complex element parsing. Because many sources come from PDFs, papers, Word or LaTeX files, and domain reports that may overlap with pretraining corpora, these benchmarks also motivate reporting provenance and de-duplication when available.

The Document-to-Markdown setting focuses on extracting holistic structured content from document images. To assess OCR capabilities in real-world scenarios, Ocean-OCR (Chen et al., 2025j) curates an evaluation dataset with 200 samples sourced from diverse English and Chinese papers, targeting three practical tasks ranging from document understanding to handwritten text recognition. As it moves towards complex full-page layouts, the field has advanced to support full-page PDF parsing. OmniDocBench (Ouyang et al., 2025b) introduces a comprehensive benchmark comprising 1.3k PDF pages, characterized by detailed block- and span-level annotations to support flexible multi-level evaluation. In parallel, olmOCR-Bench (Poznanski et al., 2025a) comprises 1.4k distinct PDF documents and enables fine-grained assessment with 7k unique

| Benchmark | Year | Task | Test Instances | Key Features |
|---|---|---|---|---|
| OmniDocBench (Ouyang et al., 2025b) | 2025 | Document-to-Markdown | 1.3k | Multi-level Annotations |
| olmOCR-Bench (Poznanski et al., 2025a) | 2025 | Document-to-Markdown | 1.4k | Unit-test Evaluation |
| Ocean-OCR (Chen et al., 2025j) | 2025 | Document-to-Markdown | 200 | Real-world Scenarios |
| OCRBench v2 (Fu et al., 2025a) | 2025 | Document OCR | 10k | Localization and Reasoning |
| KITAB-Bench (Heakl et al., 2025) | 2025 | Document OCR | 8.8k | Arabic Document Understanding |
| TableBank (Li et al., 2020a) | 2020 | Table-to-HTML | 5k | Structure Recognition |
| PubTabNet (Zhong et al., 2020) | 2020 | Table-to-HTML | 9k | Structure and Content Recognition |
| FinTabNet (Zheng et al., 2021) | 2021 | Table-to-HTML | 10.7k | Financial Domain |
| TAB2LATEX (Jiang et al., 2025b) | 2025 | Table-to-LaTeX | 5k | Unified Image Resolution |
| Table2LaTeX-RL (Ling et al., 2025) | 2025 | Table-to-LaTeX | 1.2k | Complexity Stratification |
| IMG2LATEX-100K (Deng et al., 2017) | 2017 | Formula-to-LaTeX | 10k | Dual-aspect Evaluation |
| UniMER-Test (Wang et al., 2024a) | 2024 | Formula-to-LaTeX | 23k | Multi-scenario Evaluation |
| CSFormula (Zhong et al., 2025b) | 2025 | Formula-to-LaTeX | 3k | Multi-granularity Evaluation |

Table 3: Representative benchmarks in Structured Document Parsing. We categorize these approaches by task formulation, input structure, and output language. Counts report input-level test or evaluation instances rather than training-set, collection-scale, or auxiliary unit-test sizes, and units vary across pages, documents, tables, and formulas.

test cases, covering both general patterns and challenging extraction tasks, such as tables and formulas. The progression from simple text extraction to full-page layout reconstruction reflects a growing recognition that document parsing requires evaluating not merely character accuracy but reading-order correctness and structural coherence across heterogeneous elements.

The Table-to-Code setting aims to translate tabular data into machine-readable markup, predominantly spanning HTML and LaTeX formats. For HTML-based tasks, pioneering benchmarks such as TableBank (Li et al., 2020a) leverage weak supervision from Word and LaTeX documents to facilitate table detection and structure recognition. To enable end-to-end evaluation, PubTabNet (Zhong et al., 2020) provides detailed HTML annotations for scientific tables and introduces the Tree-Edit-Distance-based Similarity (TEDS) metric for accurate structure assessment. Conversely, FinTabNet (Zheng et al., 2021) extends the target domain to financial reports, addressing unique challenges in layout and visual style using a 10.7k-table test set with cell-level annotations. More recently, the research scope has expanded to the syntactically complex domain of Table-to-LaTeX. Tab2LaTeX (Jiang et al., 2025b) pioneers this sub-domain by providing 5k compilable source code samples to specifically evaluate renderable LaTeX generation. The migration from HTML to LaTeX benchmarks is not merely a format shift because LaTeX introduces brittle compile-time validity requirements, where a single mismatched brace can invalidate the entire output. To handle varying structural complexities, Table2LaTeX-RL (Ling et al., 2025) categorizes tables into simple (<100 cells), medium, and complex (>160 cells) levels, supporting fine-grained evaluation of model capabilities in processing \multirow and \multicolumn commands. Its dual-reward design illustrates a broader evaluation concern, since textual similarity metrics like TEDS measure structural correctness but can miss visual fidelity, making rendered comparison a complementary visual-fidelity check.

Finally, the Formula-to-LaTeX setting focuses on synthesizing executable LaTeX sequences from mathematical expressions. Pioneering the field, IM2LaTeX-100K (Deng et al., 2017) establishes a robust baseline by sourcing samples from academic papers and evaluating performance via image-level pixel matching and text-level BLEU scores. To address the limitations of simple scenarios, UniMER-Test (Wang et al., 2024a) enriches the landscape by introducing 23k samples covering four representative types, ranging from simple printed to complex handwritten expressions. Recently, CSFormula (Zhong et al., 2025b) has served as a challenging benchmark for authentic scientific papers. It encompasses multidisciplinary formulas across mathematics, physics, and chemistry, and hierarchically organizes them into line-, paragraph-, and page-level categories to evaluate context-dependent generation. The progression from BLEU-based evaluation to structural edit distance and ultimately to render-based verification reflects a growing recognition that textual similarity does not guarantee executable correctness, a lesson that has since propagated to table and document parsing as well. Representative benchmarks are summarized in Table 3.

### 4.2.2 Structured Document Code Generation Methods

Methods for structured document code generation are shaped by page heterogeneity, since prose, tables, formulas, and layout relations follow different structural rules. As a result, full-page parsers balance pipeline decomposition, end-to-end VLM extraction, and RL feedback, while table and formula systems specialize around grid structure or LaTeX grammar.

Full-page document parsing exposes the pipeline-to-end-to-end tradeoff most clearly because layout routing, local recognition, and output serialization interact. The Document-to-Markdown setting has long focused on extracting structured content from document images. Traditional pipeline-based OCR models (Wang et al., 2024b; Cui et al., 2025b; Paruchuri, 2025) decompose the task into sequential stages, typically starting with layout analysis for region segmentation and subsequently employing region-specific parsers to arrange content in reading order. For example, MinerU (Wang et al., 2024b) integrates PDF-Extract-Kit (OpenDataLab, 2025) with refined pre- and post-processing strategies to improve extraction accuracy across diverse document formats. Driven by the semantic capabilities of VLMs, pipeline-based VLM methods (Feng et al., 2025; Li et al., 2025f; Cui et al., 2025a) embed VLMs into multi-stage workflows. Notably, PaddleOCR-VL (Cui et al., 2025a) combines traditional layout detection with a unified VLM for holistic content extraction. In contrast, end-to-end VLMs (Liu et al., 2025c; Poznanski et al., 2025a; Wei et al., 2025a) employ unified architectures to synthesize structured outputs directly in a single step. For instance, dots.ocr (rednote, 2025) leverages native-resolution vision encoders to achieve high-fidelity extraction. RL-based methods add a feedback layer rather than a separate parsing architecture. Infinity Parser (Wang et al., 2025a) and Logics-Parsing (Chen et al., 2025l) design verifiable rewards to capture structural consistency, while olmOCR 2 (Poznanski et al., 2025b) employs diverse binary unit tests as reward signals. Similarly, FD-RL (Zhong et al., 2025a) exploits high-entropy patterns in format-intensive content to guide RL optimization toward challenging samples.

Table recognition poses a representational challenge distinct from text extraction, as two-dimensional spatial relationships must be explicitly encoded in a one-dimensional token sequence. In parallel, the Table-to-Code domain aims to translate tabular data into machine-readable markup, evolving from visual detection to unified language modeling. Early approaches to Table-to-HTML (Zhong et al., 2020; Zheng et al., 2021) introduce attention-based frameworks that jointly perform structure recognition and cell localization. To enhance parsing accuracy, Transformer-based models such as TableFormer (Nassar et al., 2022) and VAST (Huang et al., 2023) employ end-to-end architectures that integrate object-detection decoders with coordinate-sequence modeling. UniTable (Peng et al., 2024) further unifies the task formulation by casting structure and content extraction as a single language modeling task, while SLANet (Cui et al., 2025b) combines text detection with structure prediction to handle complex borderless tables. The persistence of explicit structure prediction modules in otherwise end-to-end architectures suggests that current systems still benefit from grid-aware or structure-aware components for two-dimensional table relationships. More recently, MLLMs (rednote, 2025; Mandalm, 2025; Niu et al., 2025a) have propelled the field forward, offering exceptional flexibility in table recognition. Beyond HTML, the community also addresses the syntactically complex domain of Table-to-LaTeX. LaTeXNet (Xia et al., 2025b) achieves unified recognition through a two-stage routing architecture that dynamically directs inputs to specialized submodules. To guide precise corrections, Latte (Jiang et al., 2025b) utilizes an iterative refine-and-correct framework supported by a novel ImageEdit algorithm. Furthermore, Table2LaTeX-RL (Ling et al., 2025) proposes a dual-reward strategy, VSGRPO, that jointly optimizes structural accuracy via the TEDS metric and visual fidelity via CW-SSIM. In a broader context, OmniCaptioner (Lu et al., 2025c) establishes a unified cross-domain framework that generates precise LaTeX code for tables, formulas, and geometric figures, as well as natural scene descriptions.

Formula recognition occupies a structural middle ground, where LaTeX syntax is inherently sequential but spatial nesting and context-sensitive symbol relationships still require grammar-aware decoding that transcends pure visual recognition. Finally, the Formula-to-LaTeX setting, also known as Mathematical Expression Recognition (MER), focuses on transforming mathematical images into executable LaTeX sequences. Building upon the Transformer architecture (Vaswani et al., 2017), early pioneers like Pix2tex (Blecher, 2022) and Texify (Paruchuri, 2023) establish the standard encoder-decoder architecture. Subsequent research focuses on granular refinements to address structural challenges. Specifically, PosFormer (Guan et al., 2024) explicitly models spatial relationships via a position forest structure, while UniMERNet (Wang et al., 2024a) enhances feature extraction through detail-aware encoding. Additionally, HD-Net (Wang

et al., 2025f) resolves hierarchical complexity by employing sub-formula modules. Recent advances have introduced end-to-end models designed to balance accuracy and efficiency. PP-FormulaNet (Liu et al., 2025b) addresses this trade-off by employing dual architectures tailored to different scenarios. Conversely, DocTron-Formula (Zhong et al., 2025b) directly leverages general vision-language models without task-specific designs, effectively handling diverse granularities. This result suggests that, in some formula-recognition settings, broad visual-language pretraining can complement or exceed task-specific inductive bias. Beyond specialized architectures, Docfusion (Chai et al., 2025b) bridges the gap between continuous coordinate-based detection and discrete token-based recognition, leveraging Gaussian-Kernel Cross-Entropy Loss (GK-CEL) to enable simultaneous layout detection and content recognition within a lightweight framework.

**Scope and Trajectory.** Taken together, structured document code generation is difficult because correctness is distributed across reading order, layout structure, table grids, formula grammar, and output executability. A document can be locally readable while still losing cross-region order, table spans, formula nesting, or the syntax needed for valid Markdown, HTML, and LaTeX. This makes document generation different from ordinary OCR because the target is not only text recovery, but preservation of multiple structural grammars within one serialized representation.

Current benchmarks expose different parts of this problem, from PDF-to-Markdown reconstruction and OCR-oriented localization to table markup and formula-to-LaTeX generation. The trajectory of the field should therefore move toward more complex real-world documents with dense layouts, domain conventions, and cross-page dependencies. A realistic long-term goal is to convert diverse documents into renderable structured representations by combining the strengths of Markdown, HTML, and LaTeX as target languages. Future benchmarks and training loops should verify these code targets through rerendering or execution, while methods should use adaptive routing to decide when page context, grid-aware parsing, or grammar-aware decoding is required.

### 4.3 Academic Presentations

Academic presentation generation turns dense research content into audience-facing visual narratives, including slides, posters, and narrated presentation videos (Chen et al., 2025h). Unlike source-preserving document parsing, these tasks require models to select, reorder, and emphasize evidence while keeping the resulting artifacts visually coherent and human-editable. The central challenge is therefore to connect scientific argument structure with layout, style, and presentation flow.

#### 4.3.1 Presentations Generation Benchmarks

Academic presentation benchmarks evaluate how models turn research content into usable visual communication artifacts. Existing work covers full slide-deck generation, fine-grained slide editing, slide-to-code reconstruction, presentation-video generation, and paper-to-poster compression. We introduce these settings from slides to posters before comparing their scale and evaluation focus.

Slide-deck benchmarks differ in whether they evaluate content selection, design coherence, or generation quality over real-world decks. SlidesBench (Ge et al., 2025) and Zenodo10K (Zheng et al., 2025) provide benchmark resources at different scales for assessing design quality and coherence. Zenodo10K contains 10,448 curated presentations, but its reported generation benchmark uses 500 sampled tasks per experimental configuration. Notably, Zenodo10K uses MLLM-based metrics via the PPTEval framework to better align with human preferences. Beyond generation from scratch, practical assistants also need to modify existing decks and recover editable slide code. TSBench (Jung et al., 2025) evaluates fine-grained instruction following across text editing and visual formatting. Slide2Code (Tang et al., 2025b) assesses visual reverse engineering, while Doc2Present (Shi et al., 2025) evaluates audio-visual alignment for presentation videos.

Poster benchmarks shift the evaluation target from multi-page narrative to single-canvas compression. Paper2Poster (Pang et al., 2025) and P2PEval (Sun et al., 2025d) address information density and layout rationality through reader-simulation metrics such as the Paper Quiz and fine-grained checklist criteria. Together, slide and poster benchmarks still provide only partial evidence of rhetorical sequencing, presenter intent, and communicative effectiveness. Representative benchmarks are summarized in Table 4.

| Benchmarks | Year | Task | Test Instances | Key Features |
|---|---|---|---|---|
| *Slide Generation / Editing Benchmarks* | | | | |
| SlidesBench (Ge et al., 2025) | 2025 | Slide Generation | 585 | Domain Diversity |
| Zenodo10K (Zheng et al., 2025) | 2025 | Slide Generation | 500 tasks | PPTEval Scoring |
| TSBench (Jung et al., 2025) | 2025 | Slide Editing | 379 | Fine-grained Editing |
| Slide2Code (Tang et al., 2025b) | 2025 | Slide-to-Code | 300 | Visual Reverse Engineering |
| Doc2Present (Shi et al., 2025) | 2025 | Presentation Video | 30 | Audio-Visual Alignment |
| *Poster Generation Benchmarks* | | | | |
| Paper2Poster (Pang et al., 2025) | 2025 | Poster Generation | 100 | Paper Quiz |
| P2PEval (Sun et al., 2025d) | 2025 | Poster Generation | 121 | Fine-grained Checklist |

Table 4: Representative benchmarks for Academic Presentation Generation, classified by slide generation, slide editing, presentation video, and poster generation tasks.

### 4.3.2 Presentations Generation Methods

Presentation methods mirror the communication pipeline rather than a single model-training recipe. They follow three main routes: programmatic rendering APIs, editable object manipulation, and visual or aesthetic feedback for layout repair.

The first route uses programmatic slide representations and rendering-code interfaces. Approaches like Auto-Present (Ge et al., 2025) utilize modular libraries such as SlidesLib, which allows LLMs to focus on high-level content planning while delegating rendering details to specific API calls. Similarly, SlideCoder (Tang et al., 2025b) employs a hierarchical retrieval mechanism to reverse-engineer rendering code from images, effectively bridging the gap between visual design and code representation. The second route treats presentations as editable object structures rather than static outputs. Recent works, such as PPTAgent (Zheng et al., 2025) and Talk-to-Your-Slides (Jung et al., 2025), operate by analyzing existing templates to execute targeted modifications. This object-level view can expose editable slide elements, reducing the need to treat slides only as screenshots. The third route adds visual or aesthetic feedback to repair layout defects. PreGenie (Xu et al., 2025d) and EvoPresent (Liu et al., 2025a) introduce optimization feedback loops. Specifically, PreGenie employs a dual-review mechanism comprising code and page reviewers. In parallel, EvoPresent develops PresAesth, a multi-task RL model that integrates scoring, defect adjustment, and layout comparison to steer agents toward aesthetically superior results.

Transitioning from multi-page slides to the single-page constraints of poster generation, the critical technical bottleneck shifts toward spatial planning for content of variable lengths. PosterAgent (Pang et al., 2025) addresses this challenge through a binary-tree layout strategy integrated with a visual feedback mechanism termed the painter-commenter architecture, which dynamically mitigates text overflow issues. Alternatively, frameworks like P2P (Sun et al., 2025d) and PosterGen (Zhang et al., 2025i) adopt a decoupled paradigm that distinguishes content extraction from layout design. By deploying specialized agents acting in roles such as stylists for color optimization or curators for narrative structuring, these frameworks approximate parts of professional design workflows and aim to preserve coherent visual hierarchy during the compression of scientific content. To reduce poster generation costs, EfficientPosterGen (Tang et al., 2026) proposes the key information identification and visual-based token compression strategy as an efficiency-oriented complement to layout planning. These systems primarily address compression, overflow, and style consistency, but they still leave open whether the final poster foregrounds the intended argument and guides reader attention effectively.

**Scope and Trajectory.** Taken together, academic presentation generation is a communication-design problem rather than a source reconstruction problem. Correctness is distributed across argument selection, editable object structure, and audience attention. Slides require coherent sequential narrative, posters require single-canvas density, and narrated presentations add temporal alignment. In practice, a visually polished deck or poster can still fail through text overflow, object overlap, inconsistent style, weak claim-figure alignment, or poor audience guidance.

The trajectory of the field should therefore move toward presentation-level intermediate representations that connect claims, evidence, layout roles, visual salience, and revision history. Future benchmarks and training loops should verify claim-evidence alignment, text overflow, object overlap, visual salience, edit success, and reader recovery of the intended argument, rather than merely judging whether the artifact looks plausible.

### 4.4 Scientific Demonstration

Scientific demonstration generation asks models to produce executable visual artifacts that explain scientific mechanisms rather than only display data. The output may be a molecular script, an interactive STEM webpage, or a theorem animation, but in each case the code must remain faithful to equations, domain constraints, and pedagogical intent. This makes the task distinct from ordinary plotting because the rendered result must function as evidence-backed explanation.

#### 4.4.1 Scientific Demonstration Generation Benchmarks

Scientific demonstration benchmarks evaluate whether generated code preserves scientific meaning across the source content, executable program, rendered artifact, and validation signal. Broader scientific-agent benchmarks such as ScienceAgentBench (Chen et al., 2025p) and ScienceBoard (Sun et al., 2025c) expose adjacent workflow and tool-use requirements, while the benchmarks below focus on code-rendered visual explanation. Within this scope, the first setting is domain-constrained visual translation, where scientific diagrams must be converted into executable code with valid domain semantics. The ChemDraw benchmark (Zhao et al., 2025b) typifies this setting in computational chemistry by evaluating the translation of molecular diagrams into executable Python scripts. It measures whether models can use SMILES strings and chemistry-specific libraries such as RDKit to reconstruct complex chemical structures.

The second setting is interactive pedagogical demonstration, where the generated artifact must teach, explain, or respond to users rather than only reconstruct a static scientific object. EduVisBench (Ji et al., 2025) introduces a multi-domain framework for assessing visual reasoning in educational settings, using a fine-grained rubric informed by pedagogical theory. InteractScience (Chen et al., 2025g) evaluates scientific demonstration code generation through interactive front-end artifacts, combining Programmatic Functional Testing (PFT) with Visually-Grounded Qualitative Testing (VQT). TheoremExplainBench (Ku et al., 2025a) further extends the setting to theorem-explanation videos, testing whether Python-generated animations communicate formal reasoning in a visually intuitive and pedagogically sound manner.

#### 4.4.2 Scientific Demonstration Generation Methods

Method development in scientific demonstration generation is shaped by the need to bind scientific content to executable visual evidence. Existing methods therefore separate into renderer-driven supervision and agentic explanation workflows, but both still require validators that check whether the rendered artifact preserves the intended scientific mechanism rather than only its appearance.

Renderer-driven methods improve reliability by grounding visual artifacts in executable tools or repair loops. Chakroborti et al. (Chakroborti et al., 2025) are adjacent to demonstration generation because they improve LLM-generated scientific analysis code through data-aware prompt disambiguation, retrieval-augmented prompting, and iterative runtime-error repair. CoSyn (Yang et al., 2025f) addresses data scarcity by using code-based rendering to synthesize 400K multimodal images and 2.7M instruction-tuning samples, including chemical and circuit domains. TinyChemVL (Zhao et al., 2025e) uses RDKit rendering for molecular property optimization, while MathCoder-VL (Wang et al., 2025h) introduces FigCodifier to co-develop models and datasets through model-in-the-loop synthesis initialized from DaTikZ (Belouadi et al., 2023). These approaches make ground truth easier to obtain, but renderer-generated appearance remains only a proxy for scientific reasoning.

Agentic explanation methods instead organize scientific communication as a staged workflow. EduVis-Agent (Ji et al., 2025) coordinates specialized agents to structure learning objectives, build step-by-step reasoning, and synthesize abstract concepts into interactive learning webpages. TheoremExplainAgent (Ku et al., 2025a) moves from web interaction to video explanation by combining a Planner Agent, a Coding

Agent, and an Agentic RAG module for Manim-based theorem animations. This decomposition improves control over explanation flow, but the plan, retrieved knowledge, generated code, and rendered visualization must still be checked against the same scientific claim.

**Scope and Trajectory.** Scientific demonstration is the most validation-sensitive part of scientific visualization because visual plausibility can hide invalid mechanisms. Correctness spans numerical values, equations, domain constraints, executable code, interaction behavior, and pedagogical interpretation. A molecule can render cleanly while violating chemistry, an animation can look intuitive while proving the wrong theorem, and an interactive demo can run while exposing an unsupported relationship.

The trajectory should therefore move toward scientific demonstrations whose claims remain traceable through the code that renders them. Future systems may use domain-specific packages for interactive lessons, theorem animations, molecular structures, simulations, or research-facing explanations, but the key requirement is that equations, simulation parameters, package calls, generated code, tool outputs, and rendered states remain inspectable. Future benchmarks should combine domain validators, execution tests, simulation logs, provenance checks, and expert-facing inspection so that scientific demonstrations become traceable scientific interfaces rather than polished but unverifiable visual outputs.

> **Takeaway.** The unifying lesson is that scientific visualization code should make visual claims inspectable, not only render them. The required semantics vary by artifact, from data transformations and markup structure to argument flow, equations, domain constraints, and pedagogical intent. This shared pattern makes visual similarity insufficient, since a plausible artifact can still contain wrong values, broken structure, omitted claims, invalid mechanisms, or unverifiable tool outputs. This domain therefore motivates multi-signal validation, where rendering and execution are paired with data recovery, structural recovery, editability, provenance, domain validators, and expert inspection. Future methods should integrate domain-specific code and professional scientific packages so that generated artifacts can support analysis, communication, teaching, and research workflows.

## 5 Structured Graphics

Structured graphics move visual-code generation from pixel-level reproduction to symbolic, editable, and executable representations. Under the Section 2 taxonomy, these tasks mainly instantiate direct generation, editing, and refinement, but their correctness signals must inspect symbolic structure rather than rendering alone. Unlike GUI layouts or scientific visualization artifacts, these outputs are useful because their code exposes objects, relations, constraints, and construction procedures that can support later inspection or modification. This section reviews SVG for editable vector design, diagrams for logic and relation recovery, and CAD for parametric 3D geometry.

### 5.1 Scalable Vector Graphics (SVG)

Scalable Vector Graphics (SVG) is a path-based 2D visual-code format that describes appearance through explicit paths, shapes, groups, styles, and coordinates. This makes SVG useful for editable icons, illustrations, and UI assets, but it also makes generation difficult because models must balance rendered fidelity with compact and meaningful vector structure. SVG code generation is usually studied through NL-to-SVG, Image-to-SVG, and SVG editing. We summarize representative SVG evaluation benchmarks in Table 5.

#### 5.1.1 SVG Code Generation Benchmarks

SVG benchmarks are most clearly separated by input direction. Text- or instruction-conditioned tasks evaluate semantic alignment and design intent, while image- or reference-conditioned tasks evaluate whether visual content can be reconstructed as compact, usable vector code.

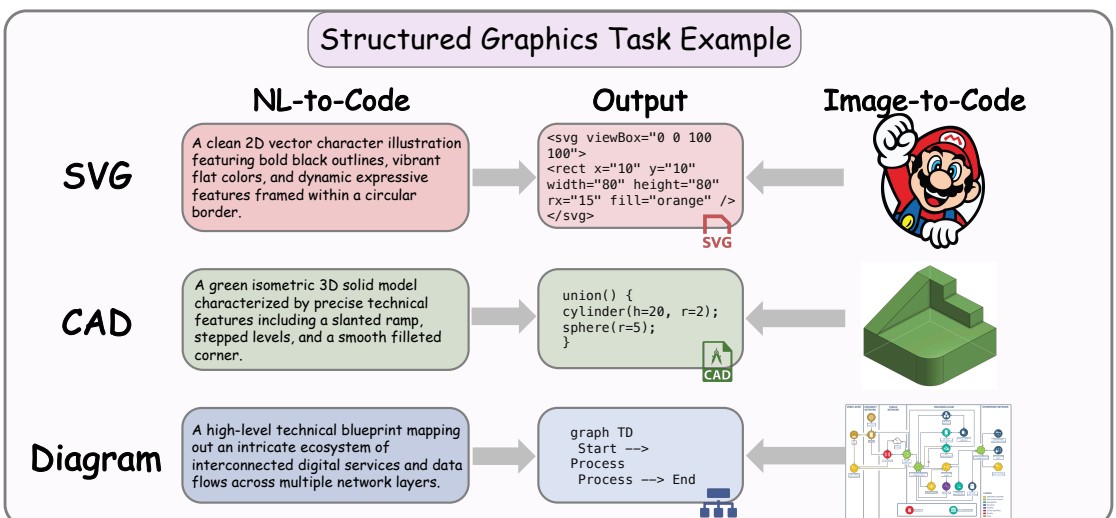

Figure 8: Examples of structured graphics generation tasks: SVG, CAD, and Diagram.

| Benchmarks | Year | Test Instances | Key Features | Primary Signal |
|---|---|---|---|---|
| *SVG Generation Benchmarks* | | | | |
| VGBench (Zou et al., 2024) | 2024 | 10.1k | Multi-format (SVG, TikZ, Graphviz) | Prompt-code alignment |
| StarVector (Rodriguez et al., 2025a) | 2025 | 24.1k | Unified Evaluation | Rendered fidelity |
| OmniSVG (Yang et al., 2025e) | 2025 | 450 | Character-Reference Gen. | Reference consistency |
| UniSVG (Li et al., 2025d) | 2025 | 2.8k | Unified Generation & Evaluation | Multi-task quality |
| SVGenius (Chen et al., 2025i) | 2025 | 2.4k | Stratified Und., Edit. & Gen. | Structural/edit signal |
| VCode (Lin et al., 2025b) | 2025 | 464 | CodeVQA & Symbolic Reasoning | Reasoning accuracy |
| Reason-SVG (Xing et al., 2025a) | 2025 | 1k | Reasoning-annotated Gen. | Reasoning/human eval |
| RLRF (Rodriguez et al., 2025b) | 2025 | 500 | Hard Image-to-SVG Recon. | Rendered reconstruction |
| RoboSVG (Wang et al., 2025g) | 2025 | 500 | Interactive SVG Completion | Completion fidelity |

Table 5: Representative SVG generation benchmarks. The Test Instances column reports evaluation or test-set size rather than training or resource scale. The primary-signal column summarizes the main property each benchmark observes rather than listing every metric.

Text-to-SVG and instruction-oriented resources focus on whether a textual prompt can be turned into semantically aligned SVG code. IconShop (Wu et al., 2023) studies text-guided icon synthesis, SVGFusion (Xing et al., 2024a) expands text-to-SVG generation to richer primitives, and LLM4SVG (Xing et al., 2025b) adds instruction-style SVG understanding and generation. VGBench (Zou et al., 2024) compares LLMs across SVG, TikZ, and Graphviz, while Reason-SVG (Xing et al., 2025a) evaluates reasoning-annotated SVG generation. UniSVG (Li et al., 2025d) and SVGenius (Chen et al., 2025i) further include text-conditioned generation, editing, and understanding within broader unified suites.

Image-to-SVG and reference-conditioned resources evaluate whether visual inputs can be converted into faithful and reusable vector programs. DeepSVG (Carlier et al., 2020) establishes an icon-scale resource for vector generation. StarVector (Rodriguez et al., 2025a) formalizes image-to-SVG evaluation through SVG-Bench. OmniSVG (Yang et al., 2025e) extends the setting to character-reference SVG generation. RLRF (Rodriguez et al., 2025b) introduces hard reconstruction settings with rendering feedback. VCode (Lin et al., 2025b) reframes natural-image understanding as symbolic SVG generation evaluated through Code-VQA. RoboSVG (Wang et al., 2025g) further broadens evaluation to image-guided and partial-input SVG completion.

Existing protocols typically combine three classes of proxy metrics. Semantic alignment metrics, including CLIP (Radford et al., 2021), BLIP (Li et al., 2022a), and SigLIP (Zhai et al., 2023), are adopted by VGBench (Zou et al., 2024), StarVector (Rodriguez et al., 2025a), and VCode (Lin et al., 2025b) to measure text-image consistency. Perceptual reconstruction metrics, including SSIM (Wang et al., 2004), LPIPS (Zhang et al., 2018), and DINOScore (Oquab et al., 2023), are used by StarVector (Rodriguez et al.,

2025a), RLRF (Rodriguez et al., 2025b), and RoboSVG (Wang et al., 2025g) to measure rendered fidelity. Task-specific structural metrics, together with edit success, code economy, and human preference studies, are used by SVGenius (Chen et al., 2025i), Reason-SVG (Xing et al., 2025a), and RoboSVG (Wang et al., 2025g) to approximate editability and perceived quality. These metrics are useful but incomplete because high scores on templated icon suites can still reflect memorized primitives rather than robust vector reasoning.

### 5.1.2 SVG Code Generation Methods

SVG methods can be organized by how they handle the discrete-continuous structure of vector code. Optimization methods keep geometry continuous and optimize through rendering, sequence models keep SVG as explicit command tokens, and LLM or RL systems add semantic planning, dataset scale, or rendered feedback.

Traditional paradigms in SVG generation primarily formulate the task as either an inverse graphics problem or a sequence modeling challenge, relying on optimization techniques and specialized neural architectures. Optimization-based methods utilize differentiable rasterizers to refine vector parameters via loss minimization. These methods assume that gradient signals from pixel-level reconstruction losses suffice to optimize both structural composition and geometric precision, an assumption that is often effective for simple shapes but can struggle with complex graphics where the parameter space is high-dimensional and non-convex. For instance, DiffVG (Li et al., 2020b) and LIVE (Ma et al., 2022) optimize parameters via loss minimization. In the realm of NL-to-SVG, VectorFusion (Jain et al., 2023) adapts Score Distillation Sampling (SDS) to optimize shape parameters. Advancing this paradigm, SVGDreamer (Xing et al., 2024b) incorporates a semantic-driven image vectorization (SIVE) process for decomposition and employs Vectorized Particle-based Score Distillation (VPSD) to enhance convergence. Conversely, Image-to-SVG approaches prioritize contour and region fidelity. SAMVG (Zhu et al., 2024a) integrates the Segment-Anything Model (SAM) (Kirillov et al., 2023) for segmentation-guided vectorization, and NeuralSVG (Polaczek et al., 2025) adopts an Implicit Neural Representation to encode SVGs with a strict hierarchical structure. In parallel, neural sequence methods learn explicit mappings for generation. Early approaches prioritize structural representation learning. DeepSVG (Carlier et al., 2020) pioneers this direction by introducing a Hierarchical VAE alongside the SVG-Icons8 dataset to facilitate non-autoregressive generation, whereas Im2Vec (Reddy et al., 2021) trains a VAE using exclusively raster supervision to generate paths without explicit vector annotations. With the adoption of Transformers, IconShop (Wu et al., 2023) establishes the core autoregressive paradigm for NL-to-SVG conversion by converting text and paths into a unified linear token sequence. SuperSVG (Hu et al., 2024a) enhances fidelity in Image-to-SVG tasks through superpixel decomposition within a multistage pipeline. The implicit assumption of sequence models is that tokenizing continuous coordinates preserves sufficient geometric precision, yet this quantization can discard fine-grained geometry unless paired with refinement or higher-level primitives.

LLM-era SVG systems address the same representation bottleneck through data scaling, domain-specific tokenization, reasoning, interaction, and rendering feedback. StarVector (Rodriguez et al., 2025a) unifies NL-to-SVG and Image-to-SVG through primitive-aware parameterization over SVG-Stack, while LLM4SVG (Xing et al., 2025b) introduces SVG-specific semantic tokens and instruction data to improve SVG understanding and generation. SVGFusion (Xing et al., 2024a), ColorSVG-100K (Chen & Pan, 2025), and OmniSVG (Yang et al., 2025e) expand coverage to richer primitives, color information, and multi-task illustration generation. A complementary line adds explicit reasoning or feedback loops. Reason-SVG (Xing et al., 2025a), SVG-Thinker (Chen et al., 2025a), and SVGen (Wang et al., 2025c) use reasoning traces to improve instruction-aligned generation, RoboSVG (Wang et al., 2025g) targets interactive completion from partial inputs, and Chat2SVG (Wu et al., 2025c) combines LLM-based templates with diffusion-based geometric optimization. RLRF (Rodriguez et al., 2025b) further introduces rendering-aware RL to compensate for the fact that autoregressive SVG models are otherwise trained largely on token-level supervision without directly observing rendered outcomes.

**Scope and Trajectory.** Taken together, SVG generation is difficult because correctness is distributed across visual fidelity, geometric precision, and structural editability. SVG code is neither a pure image representation nor an ordinary token sequence, since discrete commands define objects and paths while

continuous coordinates determine geometry. The dominant path-based representation is low-level, lengthy, and weakly aligned with semantic objects, which makes it hard for models to learn, write, and revise. This representational mismatch explains why optimization, sequence modeling, and LLM/RL methods each solve only part of the problem.

The trajectory of the field should therefore move beyond static, path-level vectorization toward LLM-friendly SVG programming frameworks. Future systems should support dynamic SVG code that can express parametric geometry, animation, interaction, and state changes, rather than only fit fixed paths to one rendered image. They should also expose higher-level primitives, structured groups, style abstractions, and editable constraints so that visual programs become easier to plan, learn, and repair. Future benchmarks should likewise separate static rendering fidelity from structural validity, dynamic behavior, edit success, and downstream reuse, making SVG outputs accountable as executable visual code rather than collections of paths.

## 5.2 Diagram

Diagrams use visual layout to encode relations such as control flow, architectural dependency, workflow state, UML semantics, and hardware connectivity. This makes diagram code generation different from SVG vectorization because the output must preserve logic, not only shapes and coordinates. Existing tasks cover both NL-to-Diagram synthesis and Diagram-to-Code translation from sketches, rendered diagrams, or domain-specific visual inputs.

### 5.2.1 Diagram Code Generation Benchmarks

Diagram benchmarks should first be separated by input direction and then by the formal structure they recover. NL-to-Diagram tasks ask whether text can be compiled into executable diagram code, while Diagram-to-Code tasks ask whether a visual diagram can be translated into a relation graph, workflow, program, or domain-specific code.

The first setting is text-to-diagram code generation. VisPlotBench (Ni et al., 2025a) incorporates Mermaid code generation tasks to assess whether natural language can be translated into executable diagrammatic code. This setting is closer to program synthesis than visual reconstruction because multiple layouts can satisfy the same textual intent, while syntax errors or missing relations can still prevent the diagram from rendering or communicating the intended structure.

The second setting is visual diagram-to-code translation, where the benchmark must recover graph logic from a rendered or sketched diagram. FC2Code (Liu et al., 2022) employs a structure recognition model to transcribe flowcharts into pseudo-code and then evaluates whether this intermediate representation can be transformed into executable code. Flow2Code (He et al., 2025) leverages rendering engines to synthesize diagram images from established codebases, curating a multilingual test set of 1.6k samples across 15 languages evaluated via Pass@k. StarFlow (Bechard et al., 2025) establishes a benchmark for translating sketch images into structured JSON workflows, comprising 2.7k samples across five visual styles and a workflow-oriented metric suite. The central benchmark bottleneck is graph-level correctness under layout ambiguity, where reversed arrows, missing branch conditions, incorrect relation types, or ungrounded symbols may be obscured by high pixel or layout similarity.

The third setting extends diagram-to-code evaluation to domain-specific structures. UML-LLaVA (Bates et al., 2025) establishes dual UML benchmarks with a large-scale synthetic set and a curated real-world set of 57 samples, facilitating evaluation across both in-domain and out-of-domain scenarios. $M^2Eval$ (Chai et al., 2025a) explores code generation from code diagrams with 300 samples across 10 programming languages. MMVG (Chang et al., 2024) extends this visual-structural paradigm to hardware design by benchmarking Verilog generation from block diagrams. Since many samples are rendered or synthesized from code or text, these benchmarks must also control synthetic leakage, where models learn renderer templates or layout priors without preserving the intended relation graph.

Across these settings, the comparison axis is therefore not dataset size alone, but target formalism: diagram syntax, workflow JSON, pseudocode, programming-language code, UML structure, or Verilog. Each target

exposes a different logical failure mode, so benchmark results should be read together with the relation type being recovered.

### 5.2.2 Diagram Code Generation Methods

Diagram methods are shaped by a relation-recovery problem: models must ground nodes, edges, labels, and topology before emitting executable code, structured JSON, diagram syntax, or domain-specific programs.

One line relies on synthetic supervision and domain-specific fine-tuning to make relation structures learnable. UML-LLaVA (Bates et al., 2025) synthesizes a large corpus of activity and sequence diagrams from randomized textual descriptions to fine-tune the LLaVA-1.5 (Liu et al., 2023b) architecture. Flow2Code (He et al., 2025) validates flowchart-to-code generation by leveraging 15k training samples derived from established codebases, bridging visual control-flow structure with executable syntax. StarFlow (Bechard et al., 2025) constructs a composite training set of 18k samples that blends synthetic graphs, hand-drawn sketches, and UI renders to generate structured JSON workflows. Draw with Thought (Cui et al., 2025c) further proposes generating mxGraph code from scientific diagrams through cognitively inspired CoT prompting.

A second line scales direct diagram generation across languages, tasks, and feedback signals. $M^2$Coder (Chai et al., 2025a) introduces $M^2$C-Instruct, a corpus spanning 50 programming languages with over 13.1M samples, to improve code generation accuracy and alignment with architectural intent. OmniDiagram (Yang et al., 2026) introduces a unified framework across diagram-to-code, text-to-code, and diagram-editing tasks, together with the VIVA mechanism, which uses generative visual interrogation as a reward signal to align rendered diagrams with code logic over a large corpus of 300k candidates. A related code-as-data direction uses diagram code to construct reasoning evaluations rather than to generate deployable diagram artifacts. FlowVQA (Singh et al., 2024) uses flowchart code to synthesize visual question-answering tasks, extending diagrammatic code representations into logic-oriented evaluation. These methods show that diagram generation benefits from SFT, synthetic data, reasoning traces, and rendered feedback, but they still need explicit validation of topology, relation labels, branch semantics, and executable constraints because visually plausible diagrams can encode the wrong logic.

**Scope and Trajectory.** Diagram code generation is ultimately a logic-compilation problem rather than a drawing problem. The basic unit is not a path or shape, but a relation among nodes, edges, labels, branches, dependencies, and typed constraints. A diagram can look clean while reversing an arrow, dropping a condition, or changing a relation type, which makes rendered similarity a weak proxy for logical correctness.

The trajectory of the field should therefore treat logic correctness as the primary target of diagram code generation. Future systems should verify whether generated code preserves graph topology, edge direction, branch conditions, typed relations, and domain constraints before optimizing layout or visual polish. Evaluation should likewise test path reachability, branch equivalence, execution behavior, and targeted visual questions, so that a visually plausible diagram cannot hide broken reasoning structure.

### 5.3 Computer-Aided Design (CAD)

CAD extends structured graphics from 2D symbolic representations to 3D parametric construction. Unlike mesh or B-rep reconstruction, code-based CAD aims to recover the operations, constraints, and feature dependencies that make a design editable. This section focuses on NL-to-CAD and CAD-to-Code, where generated representations range from serialized command sequences to high-level scripts such as CadQuery and inputs include natural language, images, sketches, point clouds, or multi-view drawings.

### 5.3.1 CAD Code Generation Benchmarks

CAD benchmarks should be organized by which part of parametric design they make observable. Early datasets test procedural command reconstruction, while newer prompt-, view-, understanding-, and repair-oriented benchmarks test whether generated CAD code satisfies design intent, compiles into valid solids, and remains editable after correction or refinement. Because many CAD datasets are introduced together with

specific methods rather than reused as independent suites, the discussion below compares each setting by representative resources, observable signals, and remaining validation gaps.

The first setting is command-sequence reconstruction. Pioneering works such as DeepCAD (Wu et al., 2021) and Fashion 360 (Willis et al., 2021) serve as widely adopted benchmarks in this category, providing approximately 8k and 1.7k samples for testing and validation, respectively. Subsequent research typically follows these established protocols, training on the designated training sets and employing the corresponding test sets for evaluation.

The second setting moves toward executable, language-conditioned, or corrective CAD code. Text2CAD (Khan et al., 2024b) employs an LLM-driven pipeline for instruction generation and filtering to construct a dataset that maps abstract CAD descriptions to detailed specifications. CADTalk (Yuan et al., 2024a) presents a CAD code-commenting benchmark with over 5.3k instances, including machine-generated and human-authored programs. SGP-Bench (Qiu et al., 2024) uses SVG and CAD code to assess the semantic consistency of symbolic graphics programs, and although it is originally an understanding benchmark, it is increasingly relevant to code generation evaluation. CADReview (Chen et al., 2025b) targets program repair by pairing erroneous CAD programs with correct reference images. ExeCAD (Niu et al., 2025c) and CADExpert (Niu et al., 2025b) further move benchmark design toward executable CadQuery generation from natural language prompts or three-view engineering drawings. These benchmarks broaden CAD evaluation beyond static geometry, but they still require separate checks for compilation, solid validity, constraint satisfaction, edit propagation, and manufacturability.

### 5.3.2 CAD Code Generation Methods

CAD methods can be grouped by the representation they ask the model to produce and the feedback they use to verify geometry. Command-sequence methods learn compact procedural histories, executable-code methods generate CAD scripts, multimodal methods map images or point clouds to construction programs, and feedback-aware methods use compilers, renderers, rewards, or reviewers to repair geometric failures. We exclude direct B-rep synthesis because this survey focuses on multimodal code generation outputs that can be serialized, executed, or edited. Across these groups, compilation success is necessary but still weaker than verified geometric validity, constraint satisfaction, and editability.

The first route represents CAD as a procedural sequence. DeepCAD (Wu et al., 2021) pioneers this formulation by using a Transformer-based autoencoder to embed CAD models and generate executable command sequences, supported by a corpus of 178k design histories. This paradigm makes the target a sequence of sketch and extrusion operations rather than an unstructured 3D surface. In NL-to-CAD, Text2CAD (Khan et al., 2024b) and CAD-Translator (Li et al., 2024b) translate textual descriptions into parametric command sequences, with Text2CAD relying on LLM-assisted annotation from beginner- to expert-level prompts and CAD-Translator aligning text with CAD operations through a cascading contrastive strategy. CAD-Llama (Li et al., 2025c) further adapts code-capable language models to CAD through hierarchical annotations that capture both structured shape information and detailed textual descriptions. This route makes CAD generation learnable as sequence prediction, but its vocabulary and ordering constraints limit complex boolean operations, assemblies, and long-range feature dependencies.

The second route increases expressiveness through executable scripts and multimodal inputs. Query2CAD (Badagabettu et al., 2024) uses proprietary LLMs such as GPT-4 Turbo to generate and refine CadQuery macros from user queries, making CAD authoring closer to interactive programming. CAD-Coder (Guan et al., 2025) produces CadQuery code with strategic planning and GRPO-based RL, explicitly optimizing both syntactic correctness and geometric plausibility. CAD-Recode (Rukhovich et al., 2025) moves reverse engineering from command sequences to Python-based CadQuery scripts by training an MLLM with a point-cloud adapter. In visual and multimodal settings, CAD-SIGNet (Khan et al., 2024a) decodes CAD commands from point clouds with Sketch instance Guided Attention, Img2CAD (Chen et al., 2025k) synthesizes commands from single images or sketches, CAD-MLLM (Xu et al., 2024a) introduces Omni-CAD with constructive modeling sequences, textual descriptions, multi-view images, and point clouds, and CAD-Assistant (Mallis et al., 2025) uses CAD-specific tools for iterative synthesis from hand-drawn sketches and 3D scans. This route broadens both the output language and the input modality, but executable syntax

and plausible visual reconstruction still do not guarantee correct dimensions, constraints, feature order, or editable design intent.

The third route makes CAD-specific execution feedback part of generation. CADFusion (Wang et al., 2025j) combines textual and visual signals by applying SFT on ground-truth parametric sequences and DPO on curated visual preference data. CAD-RL (Niu et al., 2025c) uses multimodal CoT-guided RL with CoT-based SFT for cold start and RL for post-training, while ReCAD (Li et al., 2025b) adopts a related SFT-RL pipeline with Hierarchical Primitive Learning to handle increasing CAD complexity. CAD-Judge (Zhou et al., 2025) replaces expensive VLM scoring with Compiler-as-a-Judge for preference construction and Compiler-as-a-Reviewer for test-time self-debugging, followed by a two-stage SFT and KTO training pipeline. CADReview (Chen et al., 2025b) targets automated correction by training a feedback generator and code editor to repair erroneous CAD scripts using program-image pairs. The convergence on feedback suggests a shared concern that token-level imitation alone may be insufficient when correctness depends on geometric execution, constraint satisfaction, and repair under changed design conditions.

A related shape-program line uses code to expose 3D structure beyond traditional CAD. DreamCoder (Ellis et al., 2021) and ShapeCoder (Jones et al., 2023) represent visual structures with domain-specific languages, Real2Code (Mandi et al., 2024) reconstructs articulated objects as code from segmented multi-view geometry, and MeshCoder (Dai et al., 2025) synthesizes Blender Python scripts from point clouds using a large object-code dataset built with custom Blender APIs. These works are adjacent rather than CAD-specific because they target executable shape programs more broadly, but they reinforce the same need for compiler- and geometry-aware validation.

**Scope and Trajectory.** In CAD, the central challenge is parametric correctness rather than surface reconstruction alone. A model must not only produce a shape that looks close to the reference, but also recover the construction logic that determines dimensions, boolean operations, constraints, and feature dependencies. This makes CAD different from generic 3D reconstruction because the target artifact must remain a valid design program after compilation, inspection, and later modification.

The trajectory of the field should therefore move from shape reconstruction toward verifiable parametric design synthesis for realistic engineering workflows. Future systems need compiler-aware and geometry-aware validators that test boolean operations, solid closure, constraint satisfaction, feature-tree dependencies, edit propagation, and manufacturability. Future benchmarks should include more complex parts, assemblies, multi-view drawings, revision instructions, and domain-specific manufacturing constraints, so that CAD code is evaluated as an editable engineering artifact rather than a rendered 3D surface.

> **Takeaway.** Structured graphics differ from GUI and scientific visualization because their central failure mode is structural non-equivalence. A generated SVG, diagram, or CAD model can render plausibly while losing the object hierarchy, relation graph, or parametric construction history that makes the artifact useful. The chapter therefore treats visual similarity as only an entry-level signal, since SVG outputs must remain editable vector programs, diagram outputs must preserve logical relations, and CAD outputs must preserve valid constraints and feature dependencies. This domain therefore motivates structure-aware validation, where evaluation tests editability, graph reasoning, constraint propagation, compilation or execution, and downstream reuse. Future methods should treat generated code as a symbolic visual program whose objects, relations, and constraints remain inspectable, modifiable, and semantically valid after rendering.

# 6  Frontier Tasks and Frameworks

This section turns from visual artifacts to frontier settings where code mediates perception, reasoning, and action. Unlike previous domains, the generated program is often an intermediate trace, a tool call, an environment policy, or a repair interface rather than only a final object to render. These settings correspond most directly to the programmatic tool-use and executable-policy formulations in Section 2, while unified

models span the full synthesis and acting space. We therefore organize the section around five settings that stress process reliability, including programmatic visual manipulation, video code generation, embodied control, visually grounded programming, and unified multimodal code generation.

## 6.1 Programmatic Visual Manipulation

Programmatic visual manipulation marks a shift from generating visual artifacts to using code as an executable interface for inspecting visual evidence. In the Thinking with Image paradigm (Su et al., 2025), code becomes an intermediate action space for cropping, detecting, measuring, drawing, masking, plotting, or querying an image. Under the Section 2 tool-use formulation, the main bottleneck is process faithfulness, namely whether tool code $\mathcal{C}_{\text{tool}}$ and its execution transform the input image into an answer-relevant visual state $\mathcal{I}'$ that supports the final answer $\mathcal{A}$. Most methods in this subsection correspond to Tool-Augmented Visual Reasoning, while methods that return the execution result directly as the answer approach Direct Programmatic Solving. Since this line is usually evaluated through VQA-style answer accuracy rather than dedicated code-generation benchmarks, evaluation should be read as evidence about intermediate trace validity rather than dataset scale alone.

### 6.1.1 Programmatic Visual Manipulation Evaluation Signals

Existing evaluation signals are mostly indirect. Final-answer accuracy checks whether the model produces the correct $\mathcal{A}$, trace executability checks whether the generated operation can run, and process rewards check whether the operation follows an expected tool-use pattern. None of these signals alone proves that a crop, mask, sketch, plot, memory item, or command output is causally responsible for the answer. A stronger protocol would combine answer accuracy with operation replay, region grounding, evidence ablation, and counterfactual-image tests. This is why the subsection emphasizes methods that expose intermediate operations, while treating VQA-style scores as incomplete evidence for programmatic visual reasoning.

### 6.1.2 Programmatic Visual Manipulation Methods

Existing methods follow two routes. The first route uses predefined tools, where the model controls a bounded vocabulary of APIs for OCR, detection, localization, captioning, frame retrieval, memory access, or arithmetic. The second route uses generative code reasoning, where the model writes task-specific programs to create new crops, masks, sketches, plots, measurements, or formal constructions. The former provides bounded and inspectable traces, while the latter expands the space of possible visual operations but makes relevance and faithfulness harder to verify.

Predefined-tool systems first make visual reasoning executable by turning the model into a controller over expert modules. MM-React (Yang et al., 2023) routes language-model decisions to vision experts such as OCR and detection, VipAct (Zhang et al., 2024c) coordinates vision and captioning agents for fine-grained perception, and Hydra (Ke et al., 2024) uses an RL-based controller to select reasoning paths. These systems are effective when the required evidence can be decomposed into known operations, but their tool vocabulary also determines what the model can inspect. If the decisive evidence requires an unanticipated crop, spatial construction, or unusual operation composition, an otherwise valid tool trace may miss the relevant region.

Dynamic visual streams extend the same tool-use idea from selecting spatial operations to selecting temporal evidence. In this subsection, these systems are relevant because they use tool calls to locate answer-supporting evidence in video rather than to synthesize video artifacts. DoraemonGPT (Yang et al., 2024g) schedules spatial-temporal reasoning tools with MCTS, TraveLER (Shang et al., 2024) iteratively traverses and evaluates video frames, and MoReVQA (Min et al., 2024) separates video parsing from reasoning through external memory. Video Agent methods further build tool-queryable memory or iteratively compile crucial video information through VLMs (Fan et al., 2024; Wang et al., 2024c), while VTimeCoT (Zhang et al., 2025d) uses progress bars and highlighted moments to expose temporal progression. These methods improve access to temporal context, but final-answer accuracy still cannot show whether the selected frame, memory entry, or highlight is causally responsible for the answer.

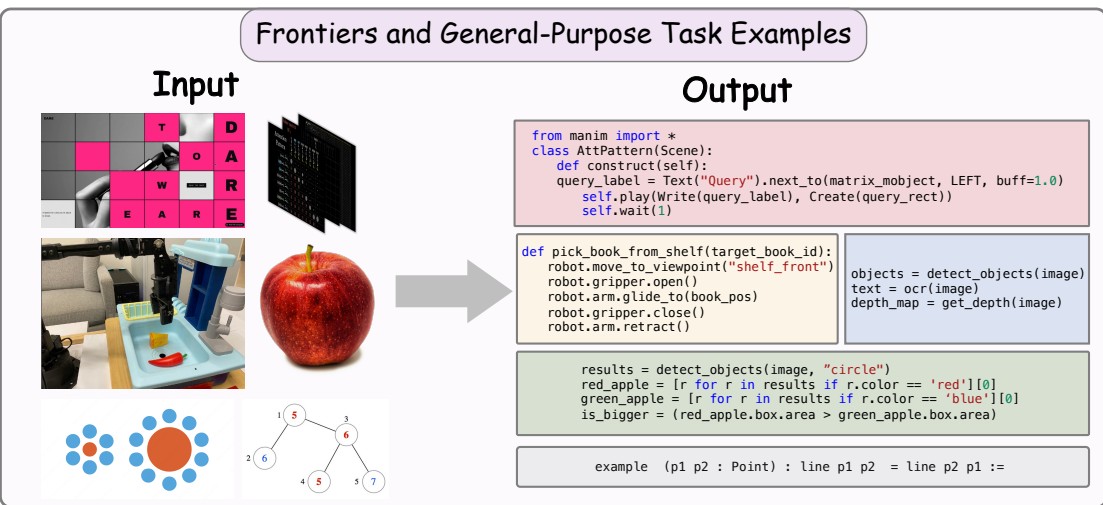

Figure 9: Tasks in the Frontier Tasks and Frameworks section, including programmatic visual manipulation, video code generation, embodied control, visually grounded programming, and unified frameworks.

A further response is to train or reward the tool-use process itself. MLLM-Tool (Wang et al., 2025b) improves API selection through instruction tuning, CogCoM (Qi et al., 2025) internalizes visual manipulation as chain-of-manipulation reasoning, Pixel Reasoner (Wang et al., 2025e) rewards pixel-space operations, and VISTA-R1 (Lu et al., 2025b) scales interleaved reasoning and tool execution in standardized environments. Compared with pure prompting, these methods supervise intermediate operations more directly. However, they still need evaluation signals that distinguish an operation that is executable from an operation that actually exposes answer-relevant visual evidence.

Generative code reasoning addresses the coverage limit of fixed tools by letting the model synthesize the visual operation itself. Early systems make compositional VQA executable through program synthesis. VisProg (Gupta & Kembhavi, 2023) translates instructions into visual-module programs, ViperGPT (Surís et al., 2023) composes vision-and-language APIs into Python subroutines, and CodeVQA (Subramanian et al., 2023) connects visual primitives with conditional logic through code. Their value is inspectability, since the reasoning structure becomes explicit. Their risk is unfaithfulness, since a valid program can still route irrelevant evidence into otherwise plausible logical steps.

Later methods use code not only to call modules, but also to construct new visual views for reasoning. Visual Sketchpad (Hu et al., 2024b) renders auxiliary sketches, ViLaSR (Wu et al., 2025b) draws spatial indicators, PyVision (Zhao et al., 2025a) synthesizes multi-turn analysis programs with external libraries, and ReFocus (Fu et al., 2025c) highlights or masks image regions for multi-hop reasoning. This shift is important because the model can create intermediate evidence when the original image is difficult to inspect directly. It also raises the verification burden, because a sketch, mask, or highlight may look plausible without isolating the evidence that determines the answer.

Recent SFT and RL-based systems therefore target the intermediate process rather than only the final answer. Skywork-R1V4 (Zhang et al., 2025g) learns from planning-execution trajectories, while Visual-ARFT (Liu et al., 2025d), Thyme (Zhang et al., 2025f), and CodeVision (Guo et al., 2025c) use execution or process rewards to encourage reliable image operations. CodeV (Hou et al., 2025) makes this motivation explicit by rewarding tool use that is both executable and evidence-consistent. DeepSketcher (Zhang et al., 2025b) similarly frames visual reasoning as active image interaction rather than text-only chain-of-thought. Together, these works expose the same failure mode, where executable or format-correct tool calls can satisfy process templates without exposing answer-relevant visual evidence.

Mathematical and geometric tasks show when generative visual operations become easier to verify. CodePlot-CoT (Duan et al., 2025) generates executable plotting code as a visual aid for mathematical reasoning, while Geoint-R1 (Wei et al., 2025b) constructs auxiliary geometric elements with Lean4-style formalization.

| Benchmarks | Domain | Test Instances | Reported Eval. Signal | Underchecked Aspect |
|---|---|---|---|---|
| MMMC (Chen et al., 2025n) | Code-to-Video | 456 topics | Teaching quality | Transition fidelity |
| PresentEval (Shi et al., 2025) | Code-to-Video | 30 documents | Content alignment | Fine-grained timing |
| Video2Code (Xie et al., 2025) | Video-to-Code | 115k triplets | Task success | Motion/contact dynamics |

Table 6: Representative video-code benchmarks. The last two columns summarize evaluation signals reported or implied by the benchmark setting and the temporal aspects that remain underchecked.

These settings provide external structure for checking the intermediate artifact, which makes the trace more verifiable than a purely plausible mask or sketch.

**Scope and Trajectory.** The trajectory of programmatic visual manipulation should move from isolated visual tools toward native code-agent workflows, but only when those workflows produce or validate answer-relevant visual evidence. As code agents become stronger, future systems can move beyond Python snippets and fixed visual APIs to operate through terminal commands, file operations, external libraries, runtime logs, and broader code-agent scaffolds. These operations remain in scope when they create a new visual state $\mathcal{I}'$, inspect a visual relation, or verify an operation trace over the input. This expansion increases coverage because the agent can construct task-specific visual operations instead of relying only on a small preset vocabulary. It also sharpens the same verification problem, because richer tool access only helps when each operation changes the available evidence in a way that supports the next reasoning step.

Future benchmarks should therefore inspect the visual interventions themselves rather than final answers alone. The key test is whether a generated crop, mask, sketch, plot, command output, or memory item targets the visual region or relation claimed by the reasoning trace. Future methods should learn visual-action abstractions that are executable, inspectable, replayable, grounded in visual regions, and tied to answer-relevant evidence. Section 7 generalizes this subsection-specific concern into cross-domain evidence logs for agentic systems.

### 6.2 Video Code Generation

Video generation in this survey refers to code-mediated video tasks, where programs either produce temporally ordered visual artifacts or recover procedures from video demonstrations. The intersection of video and code has evolved from early efforts in extracting program logic from visual demonstrations (Sun et al., 2018) into two main tasks: Code-to-Video generation and Video-to-Code synthesis. In Code-to-Video, code serves as an authoring scaffold for layouts, keyframes, narration, and transitions. In Video-to-Code, code abstracts dynamic demonstrations into executable procedures or policies. In Section 2 terms, Code-to-Video is a direct generation or refinement problem over rendered temporal artifacts, while Video-to-Code becomes an executable-policy problem when the recovered code drives actions. This subsection focuses on video as a temporal visual specification. The downstream reliability of robot execution is discussed in Embodied Control. Both directions expose the same bottleneck: discrete programs must approximate temporal dynamics that unfold continuously across frames.

#### 6.2.1 Video Code Generation Benchmarks

The central benchmark bottleneck is temporal consistency. Code can specify scenes, prompts, and procedures, but current evaluations often score end outcomes rather than whether timing and state changes are faithfully preserved. Existing video-code benchmarks split along two non-symmetric goals. MMMC and PresentEval evaluate communicative videos generated from code-like authoring structures, while Video2Code evaluates executable procedures recovered from videos. The observable evidence also differs. Code-to-Video benchmarks rely on teaching quality, content fidelity, model-based judgment, or user study, whereas Video-to-Code relies on task success after executing recovered policy code. This split matters because these signals test usefulness or completion more directly than state-trajectory fidelity.

In the Code-to-Video direction, MMMC (Chen et al., 2025n) evaluates educational videos whose scripts encode objects, explanations, and scene progressions through TeachQuiz and model-based judgment. PresentEval (Shi et al., 2025) evaluates document-to-presentation videos where generated plans must align slide

content, narration, and visual timing through content fidelity and user study. These metrics are appropriate for communicative usefulness, but they treat temporal quality indirectly. A video may preserve the right content while still using awkward pacing, weak synchronization, or discontinuous visual transitions.

In the Video-to-Code direction, Video2Code (Xie et al., 2025) provides 115k video-code-observation triplets for recovering policy programs from manipulation demonstrations. Its success-rate evaluation tests whether recovered code can complete a task, but task completion is not the same as motion fidelity. A policy can succeed while discarding velocity, contact timing, or recovery behavior that was present in the source video. We provide details about current video code generation benchmarks in Table 6.

### 6.2.2 Video Code Generation Methods

Methodologically, video-code systems use code in two different ways. Code-to-Video methods use programs as authoring structures for scenes, narration, and transitions. Video-to-Code methods use programs as compressed procedures extracted from demonstrations. The two directions share the same difficulty: code gives temporal structure, but it usually represents time as discrete steps, key states, or subgoals rather than continuous state evolution.

In Code-to-Video, executable scripts reduce the uncertainty of pixel-level generation by fixing high-level structure. Code2Video (Chen et al., 2025n) synthesizes Python scripts for educational videos, PresentAgent (Shi et al., 2025) segments documents and synchronizes visual assets with narration, and Theorem ExplainAgent (Ku et al., 2025b) retrieves and generates explanatory animations for scientific theorems. These systems show why code is useful as a planning layer because it can specify objects, layouts, step order, and narration alignment. However, the smoothness of transitions and the continuous motion between key states are still largely delegated to the rendering pipeline.

In Video-to-Code, the goal is instead to compress visual demonstrations into executable strategies. RoboPro (Xie et al., 2025) uses VLMs and Code LLMs to synthesize robotic manipulation policies from large-scale video data, reducing the need for expensive robot trajectory collection. In this subsection, RoboPro is treated as video-to-program abstraction rather than as a full account of robot deployment. This abstraction is powerful when the demonstration can be represented as ordered subgoals, but it is less reliable when success depends on continuous motion details. JanusCoder (Sun et al., 2025b) broadens the interface by covering static visual tasks and dynamic code-driven videos such as Manim animations. Its unified framing is useful, but it also makes the temporal mismatch more visible: presentation videos and robotic policies both use code, yet they encode different kinds of time.

**Scope and Trajectory.** Because current video-code benchmarks are still sparse, the trajectory below should be read as an evaluation expansion rather than an established empirical trend. Video code generation should move from sequencing visual states toward modeling time-aware state evolution. Code-to-Video systems already show why scripts are useful for prompt compliance because they make objects, layouts, keyframes, camera changes, and narration-aligned transitions explicit. However, this strength should not be conflated with full temporal control. The code often defines sparse anchors, while interpolation, pacing, easing, and perceptual smoothness are handled by the rendering engine. Thus code can improve global organization without fully exposing the temporal dynamics that make motion coherent across frames. Video-to-Code exposes the same abstraction gap from the input side. Demonstrations can often be compressed into ordered subgoals, but they become harder to represent when success depends on velocity, acceleration, contact timing, force response, or recovery from unexpected motion. Future systems should therefore expose time-aware program abstractions, including state trajectories, transition constraints, synchronization relations, and task-relevant motion dynamics.

Current metrics are meaningful but incomplete because they can verify teaching quality, content fidelity, user preference, or task success without checking whether time evolves correctly. An educational video may teach the right concept while containing awkward transitions, and a manipulation policy may complete a task while discarding the dynamics of the demonstrated motion. Future video-code benchmarks should evaluate end outcomes together with trajectory consistency, event timing, narration-visual synchronization, motion smoothness, and preservation of task-relevant dynamics.

### 6.3 Embodied Control

Embodied control instantiates the Executable Policy formulation in Section 2, where generated code $\mathcal{C}_{\text{policy}}$ or a code-based policy $\pi$ maps visual observations and high-level goals to environment actions. This subsection does not attempt to survey all embodied VLM or vision-language-action policies. Instead, it focuses on code-centric settings where programs, reward functions, specifications, or generated policy scripts mediate embodied execution. Unlike programmatic visual manipulation, the code is not only an evidence-inspection trace. It must interface with sensors, controllers, object geometry, and physical feedback. The central bottleneck is physical grounding because code is discrete and symbolic, while embodied execution is continuous, stochastic, and constrained by robot morphology, calibration, contact, and safety.

#### 6.3.1 Embodied Control Benchmarks

Embodied benchmarks make this grounding problem observable through executable environments, skill programs, and task outcomes. Early environments focus on whether abstract activities can be represented as executable action programs. VirtualHome (Puig et al., 2018) establishes this infrastructure by modeling household activities as programs for long-horizon interaction. Code-centric benchmarks then test whether language models can generate the control logic itself. Code as Policies (CaP) (Liang et al., 2022) introduces RoboCodeGen with 37 tasks that assess spatial-geometric reasoning and hierarchical control-flow synthesis. OctoVerse (Yang et al., 2024b) expands evaluation across photorealistic interiors, Minecraft, and GTA-V for vision-dependent function calls, while STEVE-21K (Zhao et al., 2024) provides 21k multimodal pairs of vision-environment data and skill-code triplets for open-world interaction. These benchmarks differ in what they observe. They test program validity in abstract activities, code-level decomposition in robot tasks, vision-conditioned function calls in simulated worlds, and skill-code alignment in open-world settings. Together, they connect high-level language understanding with evaluated environment outcomes, but most current protocols expose task outcomes more clearly than structured failure traces, safety violations, or recovery behavior. Simulator-centered protocols can also encode environment-specific affordances, embodiment assumptions, and binary rewards that overstate transfer to robots with different sensors, morphologies, or contact dynamics.

#### 6.3.2 Embodied Control Methods

Embodied methods use code according to what part of the policy stack it controls. The core code-generation route emits action scripts, skill programs, or geometric procedures that directly structure execution. Code as Policies (Liang et al., 2022) and ProgPrompt (Singh et al., 2023) use programmatic structures to expose API calls, assertions, and hierarchical control logic, making generated behavior easier to inspect than an end-to-end policy. STEVE (Zhao et al., 2024) decomposes open-world intent into granular guidelines for long-horizon environments. EmbodiedCoder (Lin et al., 2025c) uses geometric parameterization to synthesize trajectories from object point clouds, and RoboScript (Chen et al., 2024a) generates structured Python scripts for free-form robot tasks across simulation and real platforms. A related specification route generates reward terms or auxiliary constraints that support a downstream policy rather than fully instantiate it. VLM-CaR (Venuto et al., 2024) converts visual-language judgments into executable dense reward functions, turning semantic task progress into an optimization signal. This distinction matters because only the first route directly matches Section 2's executable-policy interface, while the second route supplies policy supervision or diagnostics. These systems make embodied intent more inspectable, but their reliability depends on controller APIs, camera-robot calibration, contact uncertainty, and whether execution feedback can repair a plan rather than merely report failure.

The second role is learning or feedback-based optimization, where robotic data or environment rewards are used to adapt the code-generation process. This route addresses a limitation of pure prompting because physical dynamics are difficult to infer from language and vision alone. PACT (Wei et al., 2023) learns from sensorimotor sequences through a perception-action causal transformer. RoboCodeX (Mu et al., 2024) uses multimodal post-training and iterative SFT to improve reasoning about physical constraints and motion preferences. Robotic Programmer (Xie et al., 2025) scales policy construction by translating large video demonstrations into reusable code procedures, reducing dependence on expensive robot trajectory collection.

Octopus (Yang et al., 2024b) further applies RL with Environmental Feedback, training on simulator success and failure signals. This route brings generated code closer to evaluated environment outcomes, but binary environment feedback can still hide unsafe intermediate actions, weak recovery behavior, or policies that only work for one simulator or robot body.

**Scope and Trajectory.** The trajectory of embodied code generation should move from programmatic task execution toward verifiable intent specification plus physically grounded control. Code is useful because it can express goals, object references, spatial constraints, skill composition, reward terms, and recovery conditions in a form that is easier to inspect than a black-box policy. However, code should not be treated as the entire control policy. A generated program can specify what should happen, but low-level controllers and closed-loop feedback determine whether it can happen safely under continuous motion, contact, occlusion, and sensor noise. The key design question is therefore where to place the boundary between symbolic planning and continuous control.

Future benchmarks should therefore test the boundary between symbolic intent and continuous control rather than only task completion. Embodied evaluation should expose whether failures arise from the generated specification, controller execution, contact timing, safety limits, recovery behavior, or changes in cameras, object poses, tools, and robot morphology. This keeps code interpretable as a task and constraint interface while leaving continuous adaptation to robot controllers. Section 7 discusses the broader trace protocol needed to make such failures replayable across agentic visual-code systems.

### 6.4   Visually Grounded Programming

Visually grounded programming studies code generation tasks where the visual input specifies program-relevant constraints that are difficult to fully articulate in text. Unlike visual artifact generation, the output is often an executable program or a repository patch whose correctness depends on whether the model uses diagrams, screenshots, visual examples, or rendered failures as grounding evidence. This subsection therefore exposes a compression bottleneck in the Section 2 direct-generation and refinement formulations, where visual context is often converted into textual summaries before code synthesis although the decisive constraint may lie in spatial relations, graph topology, UI state, or visual failure evidence. In agentic repair settings, the same evidence can also enter an observation-action loop where screenshots, browser traces, and terminal outputs guide patch decisions. The challenge is not only to perceive the image, but to preserve the part of the image that changes the generated code.

#### 6.4.1   Visually Grounded Programming Benchmarks

Existing benchmarks fall into two settings according to how visual evidence constrains the target code. The first setting is visually grounded algorithmic programming, where the image is part of a self-contained programming specification, execution target, or reverse-engineering target. MMCode (Li et al., 2024a) curates 3.5k online-judge problems with 6.6k images, making visual information part of competitive-programming problem statements, although many images still serve as supplementary illustrations. HumanEval-V (Zhang et al., 2024a) tightens the setting with 253 tasks where the image is indispensable and textual descriptions are deliberately minimized. ScratchEval (Fu et al., 2025b) uses block-based Scratch programs to test logical and spatial perception, while TurtleBench (Rismanchian et al., 2025) evaluates whether models can reproduce graphics through Python turtle code. Code-Vision (Wang et al., 2025d) further introduces a reverse-engineering setting, where models synthesize executable programs from algorithmic and mathematical flowcharts. These benchmarks move from visual presence toward visual necessity, but they still need ablations or counterfactual images to show that the generated program actually depends on the visual input.

The second setting is visually grounded software engineering, where visual artifacts help reproduce, localize, or verify implementation failures. SWE-bench (Jimenez et al., 2024) provides the repository-level foundation, but visual cases account for only a small fraction of the original benchmark. OmniGIRL (Guo et al., 2025b) broadens the setting with multilingual and multimodal repository issues that include buggy code and runtime screenshots. CodeV (Zhang et al., 2025e) filters 133 repository tasks where visual cues are explicitly required, reducing text-only solvability. SWE-bench MM (Yang et al., 2024e) compiles 617 JavaScript tasks from 17

| Benchmark | Test Instances | Task Type | Language | Visual Role | Correctness Signal |
|---|---|---|---|---|---|
| MMCode (Li et al., 2024a) | 3.5k | Code generation | Python | Problem-statement images | Program correctness |
| HumanEval-V (Zhang et al., 2024a) | 253 | Code generation | Python | Indispensable visual specification | Code pass rate |
| Code-Vision (Wang et al., 2025d) | 338 | Code generation | Python | Flowchart-to-code grounding | Executable logic |
| ScratchEval (Fu et al., 2025b) | 305 | Visual reasoning | Scratch | Block-program visual reasoning | Answer accuracy |
| TurtleBench (Rismanchian et al., 2025) | 260 | Visual execution | Python | Rendered graphics target | Render match |
| OmniGIRL (Guo et al., 2025b) | 959 | Software repair | Multi | Runtime screenshot evidence | Issue resolution |
| Visual SWE-bench (Zhang et al., 2025e) | 133 | Software repair | Python | Required visual repair evidence | Patch success |
| SWE-bench MM (Yang et al., 2024e) | 617 | Software repair | JavaScript | Screenshot/video repair evidence | Patch success |

Table 7: Summary of benchmarks for visually grounded programming tasks. We feature them according to task type, output code language, the role of visual evidence, and the correctness signal used for evaluation.

repositories with visual scenarios such as interactive mapping and web rendering. This progression moves from incidental visual artifacts toward tasks where images and videos are part of the repair evidence. These benchmarks better approximate real development, yet final pass rates can still be weak evidence for grounding if repository context or text-only issue descriptions allow shortcuts. A comparison is shown in Table 7.

### 6.4.2 Visually Grounded Programming Methods

Methods in this area follow two routes. The first route converts visual inputs into textual or symbolic surrogates before code synthesis. This design reflects a practical asymmetry because VLMs can describe visual content, while specialized Code LLMs are usually stronger at producing executable programs. Code-Vision (Wang et al., 2025d) converts flowcharts into Mermaid code before generating target programs, reducing compilation failures by giving the Code LLM a structured intermediate representation. HumanEval-V (Zhang et al., 2024a) separates VLM-based description from Code-LLM synthesis, while CodeV (Zhang et al., 2025e) converts visual inputs into fine-grained descriptions and structured summaries. This strategy works when diagrams, examples, or screenshots can be compressed into language without losing the program-relevant constraint. It is reliable for symbolic schemas and explicit labels, but weaker when geometry, topology, visual grouping, or transient UI state carries information that a textual surrogate cannot preserve.

The second route uses visual feedback inside software-engineering agents. SWE-agent (Yang et al., 2024c) establishes an agent-computer interface for repository editing, and SWE-agent M (Yang et al., 2024d;e) extends this interface with browser interaction, screenshotting, image viewing, and terminal operations so that agents can reproduce visual issues and verify fixes. Agentless (Xia et al., 2024) uses hierarchical localization from files to methods and lines before patch generation, while AutoCodeRover (Zhang et al., 2024b) emphasizes automated repository search and repair. These are general repair infrastructures rather than visually grounded methods by themselves, and their visual relevance appears only when localization, reproduction, or validation depends on screenshots or browser feedback. GUIRepair (Huang et al., 2025) makes this loop explicit through an Image2Code module that generates reproduction scripts and a Code2Image module that captures screenshots after executed fixes. By comparing post-patch screenshots with issue images, the system creates a visual repair signal. This evidence is reliable only when failures are reproducible, localization is correct, and post-patch checks cover nearby states rather than only the visible symptom.

Across both routes, the bottleneck is not perception alone. The issue is whether visual evidence survives the path into code generation or patch refinement. Textual surrogates can make code generation easier, but they risk losing spatial and state information. Agentic feedback can make repair more grounded, but it depends on reproducible environments, stable screenshots, and meaningful post-patch tests.

**Scope and Trajectory.** The trajectory of visually grounded programming should move from textual compression toward visual evidence as a first-class programming constraint. In the Section 2 direct-generation setting, the visual input is part of the specification for $\mathcal{C}_{gen}$. In the refinement setting, it is evidence for deciding whether $\mathcal{C}_{draft}$ should become $\mathcal{C}_{refined}$. Textual summaries, Mermaid conversions, captions, and structured descriptions are useful because they route visual information into stronger code generators. Their limitation is that they can flatten spatial layouts, graph topology, UI state, and visual failure evidence into incomplete language. Future systems should therefore preserve richer visual structures during code synthesis, including graph edges and nodes for flowcharts, DOM and browser state for UI screenshots, rendered failure states, and interaction traces that remain connected to the generated program or patch.

Future benchmarks should evaluate whether visual evidence changes the generated program in the expected place, rather than only whether the final program passes. Shortcut control is especially important once benchmark patterns become familiar, because text-only statements or repository context can allow a model to pass without using the image. For software engineering, screenshots, browser traces, terminal logs, reproduction scripts, localized edits, and post-patch executions should remain linked enough to show which visual failure motivated which code change. This would make visually grounded programming a test of evidence-conditioned code generation rather than a text-only programming task with optional images.

## 6.5 Unified Multimodal Code Generation

Unified multimodal code generation asks whether the visual-code capabilities reviewed in previous sections can be served by shared models and shared representations rather than by isolated domain systems. The goal is not only broader task coverage, but candidate shared visual-code primitives that could support Section 2 synthesis, editing, refinement, tool-use, interactive artifacts, OCR, visualization, and software tasks within a common interface. The central tension is a generalization paradox. Adding more domains increases coverage, but it does not by itself prove that a model has learned abstractions that transfer across tasks.

### 6.5.1 Unified Multimodal Code Generation Benchmarks

Unified benchmarks should be read as different tests of visual-code grounding rather than as larger domain collections alone. One group evaluates reconstruction or extraction, where the target is structured code recovered from visual input. A second group uses rendering code to generate synthetic multimodal data. A third group evaluates interactive artifacts, visually grounded programming, or iterative refinement. This diversity is necessary for unified evaluation, but it also makes direct score comparison unreliable unless reports specify which Section 2 objective is being tested and how leakage, saturation, and metric agreement are controlled.

For reconstruction and generated-data settings, Image2Struct (Roberts et al., 2024) assesses whether VLMs can extract structured code, such as LaTeX and HTML, from visual images across webpages, mathematical formulas, and musical scores. To evaluate visual fidelity, Image2Struct introduces metrics including Cosine Inception Similarity (CIS) and Earth Mover Similarity (EMS). Similarly, CoSyn (Yang et al., 2025f) adopts a code-first paradigm, leveraging LLMs to synthesize rendering code across 9 image categories before constructing QA pairs. These benchmarks show how rendering code can bridge visual and textual modalities, but their signals still emphasize reconstruction and generated-data utility.

For dynamic settings, ArtifactsBench (Zhang et al., 2025a) evaluates visual-interactive artifacts with 1.8k queries across nine domains and uses a checklist-guided MLLM-as-Judge pipeline to verify executability and interaction logic. InfiBench-V (Jiang et al., 2025a) targets real-world applicability with 322 visually rich questions where images are indispensable, covering 13 programming languages across front-end, back-end, data science and machine learning, mobile and desktop development, and IT operations. VisPlotBench (Ni et al., 2025a) focuses on NL-to-Visualization agents with 888 tasks across eight programming languages and a multi-round self-debug protocol for iterative refinement. Together, these benchmarks broaden the interface, but they also expose why unified scores must report which correctness signal is being measured.

### 6.5.2 Unified Multimodal Code Generation Methods

Method development in unified multimodal code generation starts from a practical mismatch: most visual-code models still target a single task or narrow scenario, while real-world use requires broader coverage over visual inputs, code targets, interaction states, and execution environments. Early foundational works therefore build cross-domain data and representation infrastructure. GOT (Wei et al., 2024) expands OCR beyond plain text to chemical, chart, and geometric scenarios through a three-stage training paradigm, while BigDoc (Rodriguez et al., 2024) provides a large-scale open-source dataset for multimodal document and code-centric tasks. Recent systems push this idea further through SFT-scale mixtures and model integration. VisCoder2 (Ni et al., 2025a) introduces VisCode-Multi-679K for visualization generation and correction, VisCodex (Jiang et al., 2025a) merges a Code LLM with a VLM backbone and introduces MCD-598k for multimodal coding tasks, and JanusCoder (Sun et al., 2025b) integrates text- and vision-centric tasks with

JanusCode-800K. These systems reduce interface fragmentation, but their open question is whether broader mixtures induce reusable visual-code primitives or mainly improve task acceptance.

A second route adds feedback-aware optimization to move beyond SFT alone. VinciCoder (Zhao et al., 2025b) uses coarse-to-fine visual similarity as a reward signal, while OCRVerse (Zhong et al., 2026) unifies OCR and programmatic tasks in an end-to-end VLM with decoupled textual and visual rewards for RL optimization. Across these systems, unification is mostly operationalized through data mixtures, model integration, and feedback signals. A stronger evaluation criterion is whether these ingredients produce controlled transfer across visual-code primitives rather than only broader benchmark coverage. Future research should therefore prioritize data-efficient training, explicit primitive sharing, and execution-aware validation instead of treating larger domain mixtures as sufficient evidence of unification.

**Scope and Trajectory.** The trajectory of unified multimodal code generation should define unification by measurable transfer rather than by dataset aggregation alone. A model that accepts many task formats is not necessarily a model that shares visual-code abstractions across tasks. The unresolved assumption is that training on many code-image-instruction tuples will induce reusable notions of axes, panels, nodes, text regions, layout hierarchy, events, and state change. Current reports more often validate broad task acceptance than controlled primitive-level transfer. This subsection therefore identifies the domain-specific limitation: unified systems may route inputs to task-specific behaviors without learning shared mechanisms.

Future unified systems should therefore report not only aggregate task coverage, but also whether shared representations improve or harm specialized syntax, layout, interaction, and domain constraints. The broader protocol for testing held-out transfer and metric agreement is discussed in Section 7; here the key point is that unification remains unproven until shared visual-code mechanisms can be separated from ordinary in-distribution task acceptance.

---

**Takeaway.** The frontier settings push multimodal code generation beyond the artifact-centered problems studied in the previous sections. GUI generation tests interaction behavior, scientific visualization tests domain meaning, and structured graphics tests editable symbolic structure. Frontier tasks add a further layer: code becomes the mechanism for inspecting evidence, modeling time, acting in environments, repairing programs, or attempting transfer across tasks. Correctness therefore depends not only on the final output, but also on whether the intermediate trace or policy actually supports the result.

Programmatic visual manipulation, video code generation, embodied control, and visually grounded programming motivate verifiable traces and state trajectories, where final-task success is paired with replay, temporal or physical diagnostics, visual ablations, and counterfactual inputs. Unified models motivate a separate transfer question, where held-out tests examine whether shared visual-code mechanisms carry across tasks rather than merely expanding task coverage.

---

## 7 Future Directions

The preceding Scope and Takeaway paragraphs point to a common verification question: after code is generated from visual input, what evidence shows that it preserves the intended visual, structural, semantic, or interactive behavior? Across the surveyed domains, code may serve as a rendered artifact, an editable representation, a tool trace, or an executable policy, and each role exposes a different validation gap. These directions ask what evidence validates generated artifacts, edited or refined code, tool-use traces, and executable policies after rendering, execution, or interaction. The four directions below organize this question around multi-signal validation for artifacts, multi-state verification for interactive and temporal systems, cross-task transfer testing for unified models, and verifiable traces for agents.

| Proxy judge | Example | Failure mode | Companion checks |
|---|---|---|---|
| Visual | MSRL (Chen et al., 2025d) (text&VLM judge), RLRF (Rodriguez et al., 2025b) (SSIM&CLIP score), VinciCoder (Zhao et al., 2025b) (DINO score). | Over-scores surface match, while missing data, structure, editability, and semantics. | Data, structure, editability, and interaction checks. |
| Text/code | Table2LaTeX-RL (Ling et al., 2025) (TEDS&CW-SSIM score), LATTE (Jiang et al., 2025b) (LaTeX edit score), Infinity Parser (Wang et al., 2025a) (layout reward), FD-RL (Zhong et al., 2025a) (format reward). | Checks local strings or syntax, while missing order, layout, relations, and execution. | Reading-order, layout, rendering, and execution checks. |
| Preference | DualDPO (Zhang et al., 2025h) (DPO preference), ReLook (Li et al., 2025e) (VLM critic), CADFusion (Wang et al., 2025j) (CAD preference). | Prompt-sensitive, biased, weakly reproducible, hard to localize, and hackable. | Judge agreement, rubric splits, human calibration, and objective tests. |
| Agent replay | Coder-CUA (Lin et al., 2025a) (CUA replay score), WebGen-Agent (Lu et al., 2025d) (GUI-agent interaction score), WebVIA (Xu et al., 2025c) (interaction-graph replay score). | Limited by agent capability, mixes code quality with policy performance, and rewards shortcuts. | Independent replay, action logs, bounded actions, and counterfactual tests. |
| Trace | Visual-ARFT (Liu et al., 2025d) (tool-trace reward), CodeV (Hou et al., 2025) (tool-use reward), Pixel Reasoner (Wang et al., 2025e) (pixel-operation reward). | Rewards plausible traces without proving causality, grounding, or faithfulness. | Evidence ablations, operation counterfactuals, and trace replay. |

Table 8: Common proxy judges and failure modes in multimodal code intelligence. Reliable evaluation generally requires a validator stack rather than a single reward.

## 7.1 Toward Multi-Signal Validation

Validation should be aligned with the role and use value of each visual-code artifact. The preceding sections show that visual similarity is a useful lower bound, but it cannot certify interaction behavior, data and scientific semantics, document structure, symbolic editability, or geometric constraints at the same time. A single reference image, reference program, or VLM preference score therefore cannot serve as a universal correctness signal. Feedback-aware methods are best read as local attempts to expose missing validators, rather than as evidence for a universal reward. MSRL makes chart feedback more structured (Chen et al., 2025d), Table2LaTeX-RL combines structural and visual signals for table markup (Ling et al., 2025), RLRF optimizes rendered SVG feedback (Rodriguez et al., 2025b), and CADFusion introduces preference signals for CAD generation (Wang et al., 2025j). Together, they show that each reward makes one property more observable while leaving other properties underchecked.

These observations motivate the proxy-judge taxonomy in Table 8, which groups common rewards and judges by their observable evidence, typical failure modes, and required companion checks. The broader goal is a diagnostic profile rather than a single scalar score. Such a profile should separate visual similarity, execution success, textual correctness, data or semantic fidelity, structural validity, editability, and interaction correctness, making it clearer whether an artifact is visually wrong, semantically wrong, structurally unusable, non-editable, or behaviorally broken. Reward design should therefore state the property being optimized, report companion validators, and distinguish training rewards from held-out reliability checks.

## 7.2 Toward Multi-State Verification

Stateful visual-code tasks should be evaluated as execution episodes rather than isolated renderings. GUI generation makes this limitation explicit because a page can reproduce a screenshot while failing under clicks, routing, resizing, or state updates. Mobile generation faces the same issue under a less transparent runtime, so current benchmarks rely on design-tool states, UI hierarchies, emulator checks, DSLs, or learned rewards as

partial substitutes for native execution. Interaction-focused web benchmarks such as Interaction2Code (Xiao et al., 2025a), MRWeb (Wan et al., 2024), and IWR-Bench (Chen et al., 2025m) show how evaluation can move from a static rendering toward executable behavior. Adjacent computer-use environments such as WebArena (Zhou et al., 2023), VisualWebArena (Koh et al., 2024), and OSWorld (Xie et al., 2024) are not primary code-generation benchmarks, but they provide useful evidence for replayable actions and task-completion protocols.

The same view applies beyond interfaces. A scientific demonstration can execute while communicating an invalid mechanism, a video script can specify plausible keyframes while losing event timing, and an embodied program can state the right goal while failing under contact, occlusion, or controller limits. Future benchmarks should therefore define an episode with initial states, generated code or actions, intermediate observations, expected transitions, validator outputs, and recovery cases. The required checks differ by substrate, including DOM and state assertions for web tasks, design-operation traces or emulator gestures for mobile tasks, synchronization checks for video tasks, and simulator or controller diagnostics for embodied tasks. The evaluated object becomes a trajectory of visual-code execution, not a visually plausible endpoint.

### 7.3 Toward Testing Cross-Task Transfer

Unified models should be evaluated by whether abilities transfer across tasks, not only by whether they accept more task formats. Systems such as JanusCoder (Sun et al., 2025b), VisCoder2 (Ni et al., 2025a), and VisCodex (Jiang et al., 2025a) expand the range of visual-code inputs and outputs. The open question is whether this breadth produces reusable visual-code skills, such as layout reasoning, symbolic relation modeling, and interaction understanding, rather than only stronger in-distribution task performance.

Future benchmarks should therefore separate task acceptance from cross-task transfer through splits over held-out skills, primitives, and compositions. A minimal protocol should compare a base mixture, a source-domain-augmented mixture, and a matched-size control mixture on provenance-filtered target tasks, then report both positive and negative transfer. Useful protocols would test whether chart training improves diagram layout reasoning, whether document structure learning improves visually grounded programming, or whether interaction supervision improves repair of generated artifacts. Results should be reported with counterfactual tests, modality ablations, and de-duplication checks, because a larger mixture can improve average coverage while weakening specialized syntax, layout, or domain constraints. This would make unified multimodal code generation a falsifiable claim about shared visual-code mechanisms, rather than a label for broad task packaging.

### 7.4 Toward Verifiable Agent Traces

Agentic visual-code systems need process-level evidence that connects visual evidence, tool use, code actions, and final outcomes. In these settings, code may create intermediate visual operations, repair a repository from screenshots and logs, or specify an embodied or temporal policy whose outcome depends on environment feedback. Systems such as Visual-ARFT (Liu et al., 2025d), the tool-use CodeV system (Hou et al., 2025), WebGen-Agent (Lu et al., 2025d), Coder-CUA (Lin et al., 2025a), and GUIRepair (Huang et al., 2025) show the value of execution and feedback, but final success alone cannot prove that the trace is faithful to the visual evidence or causally responsible for the result.

A concrete research target is an evidence log for visual-code agents. Each entry should record the observation used, the cited visual region or tool output, the code region or action changed, the validator expected to improve, the replay result, and the fallback or rollback decision when evidence is insufficient. Such logs would support replay, visual ablation, counterfactual inputs, permission control, simulator or emulator guards, and human review. They would also let evaluations attribute failures to perception, code synthesis, environment execution, validator design, or unsafe action selection, turning agentic multimodal code intelligence from a black-box success metric into a verifiable process.

# 8 Limitations

This survey is bounded by the public papers, benchmarks, and repositories available during our collection period. Some recent systems, closed-source deployments, and domain-specific tools may still be missing, so the taxonomy should be read as an organizing view rather than a final boundary of the field. Because many papers introduce their own datasets and metrics, the survey may also overrepresent benchmark-proposing works and underrepresent deployed systems without public artifacts. Closed-source model reports, private evaluation sets, and rapidly changing arXiv releases further limit the reproducibility of cross-paper comparison.

Cross-method comparison remains limited because benchmarks observe different slices of correctness. We therefore avoid a universal ranking and instead emphasize within-domain comparisons, common failure modes, and reliability risks such as leakage, benchmark saturation, and judge sensitivity. Our cross-task transfer discussion is also agenda-setting, since current evaluations rarely isolate causal transfer and still underexplore deployment-facing concerns.

# 9 Broader Impact

Multimodal code intelligence can lower the barrier to visual programming by allowing users to express intent through screenshots, diagrams, sketches, videos, or natural language, then obtain code for interfaces, charts, documents, demonstrations, SVG, CAD, or robot policies. It can also help experts turn visual feedback into executable revisions, making artifacts easier to inspect, edit, and reuse. The main risk is that visual plausibility can hide serious errors, including wrong chart data, lost document structure, invalid scientific mechanisms, broken interactions, insecure code, or unsafe physical actions.

Agentic systems add privacy and safety concerns when they operate browsers, files, APIs, design tools, proprietary repositories, or robots. Screenshots and design files may contain private information, generated code may leak or misuse proprietary context, and embodied policies may behave differently outside simulation. Deployment should therefore pair generation with provenance tracking, permission scopes, execution logs, domain validators, human review for high-stakes uses, rollback mechanisms, and clear separation between model suggestions and user-approved actions.

# 10 Conclusion

In this paper, we conduct a structured survey of Multimodal Code Intelligence, organizing the landscape into four domains: Graphical User Interface, Scientific Visualization, Structured Graphics, and Frontier Tasks and Frameworks. We provide an extensive review of existing benchmarks and methodologies, analyzing how models translate visual perception into diverse executable representations. Specifically, our survey covers a wide range of multimodal code generation tasks, such as interfaces, charts and documents, SVG/diagram/CAD programs, and code-mediated tool-use or embodied policies. Crucially, this work addresses a significant gap in the current literature regarding the utilization of code as an executable interface for many visual tasks. In this paradigm, executable code serves as a versatile intermediate medium that enables models to ground visual reasoning, invoke external tools, and support open-ended problems in dynamic environments. While text-based program synthesis is mature, this approach of leveraging code for visual problem-solving remains fragmented. Our analysis shows that progress depends not only on generating plausible code, but also on making visual-code artifacts verifiable through multi-signal validation, multi-state verification, cross-task transfer testing, and verifiable agent traces. The central conclusion is therefore that multimodal code intelligence should be evaluated by the evidence its code exposes after rendering, execution, interaction, and replay, rather than by visual plausibility alone. By establishing a clear taxonomy and organizing the evaluation landscape, this work provides a practical reference for studying multimodal coding systems whose outputs are not only visually plausible, but also executable, verifiable, editable, and grounded in the intended visual evidence.

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
