# OpenReview forum: "Beyond NL2Code: A Systematic Survey of Multimodal Code Intelligence"
_TMLR — Under review for TMLR_

### Review · Reviewer_79Wo · 2026-04-24

**Summary Of Contributions:**

This paper presents an extensive survey of work on multimodal code intelligence. It covers four domains: GUIs, visualizations, graphics and frontier frameworks. While each of these domains, it discusses the existing benchmarks and methods. It briefly discusses future directions on multimodal code intelligence.

Strengths
- Multimodal code intelligence has many applications and this paper organizes the existing literature based on application domains.
- The coverage of tasks within each domain is quite exhaustive.

Weaknesses
- Beyond cataloging the existing works along different application domains, the paper does not offer any original connections and data-driven insights (e.g., comparison of approaches on the same benchmarks).
- The notes on future directions (Sec 7) are specific to multimodal code intelligence, but are mostly shared with (non-multimodal) code intelligence and general trends in the research community.
- The prominently mentions the code repository. I took a look at it and found that it has only the bibliography in markdown.

**Audience:**

No

**Audience Explanation:**

The paper can serve as an entry point to find a bulk of work for someone interested in multimodal code intelligence, but beyond a well-curated bibliography and clustering by application domains, I am unsure what other value people can derive.

**Broader Impact Concerns:**

I don't particularly anticipate broader impact concerns.

**Claims And Evidence:**

No

**Claims Explanation:**

I mean there are no clear claims in this paper. It is a survey paper so I am not expecting original contributions per se. But as mentioned above, it is mainly a catalog of papers. No deeper insights are sought and presented like what works, what doesn't, why, how different methods compare with each other and so on.

**Requested Changes:**

- Seek and present deeper insights by looking into the evaluations in the presented papers and discuss their comparative advantages and disadvantages
- Even better conduct some experiments of your own and present findings
- Release some reusable code in the repository for others to experiment
- Sharpen the future work section with directions more specific to multimodal code intelligence
- Extend the scope to discuss evaluation and RL environments like visualwebarena and related works

---

> ### Author Response · Authors · 2026-06-07
> **Response to Reviewer 79Wo**
>
> We thank the reviewer for pointing out that the original submission needs stronger synthesis beyond paper categorization. In response, we revise Sections 3-6 to emphasise cross-domain validation gaps, add paper-count and publication-trend figures to Figures 1 and 3, and rewrite Section 7 (Future Directions) to focus on verification issues specific to multimodal code intelligence.
>
> **Deeper cross-domain insights.** Thanks for your advice. We revise the paper so that the organizing question is not only "which papers belong to which domain", but also "what evidence validates the generated code after rendering, execution, interaction, or replay." This leads to a recurring observation across Sections 3-6: visual similarity is useful but incomplete, and each task family requires different companion validators. Section 7 condenses the Scope and Trajectory paragraphs and Takeaway boxes into a verification-centered agenda.
>
> **Comparative advantages and disadvantages.** Thanks for your advice. We strengthen the discussion of what different methods and benchmarks can and cannot certify. Across Section 3-6, we now describe whether a benchmark observes rendered fidelity, code structure, recovered data, editability, interaction replay, scientific validity, constraint satisfaction, or agent task completion. We also discuss failure modes such as visually correct but semantically wrong charts, non-editable SVG paths, invalid CAD constraints, broken UI handlers, unfaithful tool traces, and simulator-specific embodied policies.
>
> **Quantitative evidence.** Thanks for your advice. We add Figure 1 and Figure 3 to make the survey coverage and trend more concrete. Figure 3 shows the distribution of the 282 unique papers across four domains and subdomains, and Figure 1 shows how the literature grows from 2024Q1 to 2025Q4. We agree that new experiments would further strengthen the paper. Given the scope of this revision, we instead add coverage statistics and diagnostic tables that are tied to the revised bibliography and section citations.
>
> **Evaluation and RL environments.** Thanks for your advice. We add discussion of environment-style evaluation in Section 7.2 (Toward Multi-State Verification). In particular, we now discuss interaction-focused web benchmarks and adjacent computer-use environments such as WebArena, VisualWebArena, and OSWorld as useful background for replayable actions and task-completion protocols. We also connect these environments to the broader need for multi-state verification rather than single-state visual reconstruction.
>
> **Making future work specific to multimodal code intelligence.** Thanks for your advice. We rewrite the whole Section 7 to make the future directions more specific to visually grounded code. Section 7.1 focuses on multi-signal validation for generated artifacts, Section 7.2 focuses on multi-state verification for interactive and temporal systems, Section 7.3 focuses on cross-task transfer of visual-code primitives, and Section 7.4 focuses on verifiable agent traces that connect visual evidence, tool use, code edits, and execution outcomes.
>
> **Repository usefulness.** We value the reviewer's feedback regarding the utility of our repository. We are currently maintaining and updating the repository to include the revised taxonomy, detailed benchmark/method tables, paper metadata used for trend analysis, and execution scripts for reproducing our survey statistics. We aim to ensure the repository serves as an interactive, reusable toolkit for the community rather than just a curated bibliography.

---

### Review · Reviewer_Cqr5 · 2026-05-03

**Summary Of Contributions:**

The authors propose an extensive survey paper on "Multimodal Code Intelligence", defined as systems that either generate code from visual inputs or use code for visually grounded reasoning and actions.  The main contributions are a 1) taxonomy organizing the literature into four domains (GUI, Scientific Visualization, Structured Graphics, and Frontiers Frameworks ) with several key tables as features and metrics, with, 2) task formulation  into synthesis and code-centric reasoning and acting, 3) analyzing future directions for the field.

**Strengths**

The topic is active, and for the multimodality part, currently undersurveyed to my knowledge: existing code-LLM surveys are predominantly text-centric and treat multimodality as a downstream concern; a survey organized around the visual modality of code generation is an interesting niche. The benchmark/method taxonomy is well-organized at the figure level, and the task definitions in breadth of coverage are impressive, with more than 200 works.

**Weaknesses**

My main concern is that the manuscript reads more like a categorized bibliography than a critical re-elaboration and critical analysis: the paper adds limited analytical value beyond what a careful reader of the cited primary works would already obtain. Most of the cited works are like “A and B do this, X and Y do that” without deeper analyses. Based on this, this is a list of weaknesses:

1. No quantitative cross-method comparison anywhere in the paper: while hard to do, this would add much value (what is better than what? what could be the failure modes of a method?)

2. No discussion of benchmark saturation, leakage, or metric reliability

3. The Section 2 formalism is introduced but then almost never used to organize Sections 3 to 6

4. No survey methodology section despite "systematic" in the title (i.e., what methods were used to categorize such extensive literature? To what extent were LLMs used for this?)

5. No Limitations section. What are the limitations of the current survey? What areas were underexplored?

6. No Broader Impact section -- given the impact on society of such systems and the survey nature, I would expect some discussions regarding this

**Audience:**

Yes

**Audience Explanation:**

Yes, the survey is timely, and the bibliography is accurate.

**Broader Impact Concerns:**

Societal impact of code generation tools (even a small section, given this topic has been discussed extensively already in and outside academia)

**Claims And Evidence:**

No

**Claims Explanation:**

My reason for answering “No” here is that, throughout the text, some claims are unsupported by evidence.

1. “Code constitutes the universal action space for multimodal general intelligence” (abstract). I strongly disagree with this, at least in the way it's worded. What is the evidence for this? Moreover, do intelligent beings, including humans, use “code” for general intelligence? I would argue not quite (unless one wanted to go into determinism and philosophy, which likely is not the paper’s point).

2. “Visually-grounded code generation marks a definitive breakthrough toward autonomous software agents” (abstract). Again, this is an exceptional claim in need of exceptional proof. In my experience, even non-visual code generation works quite well. This sentence also reads as an overclaim.

3. “Four pivotal technical shifts” (section 7). How are these pivotal? What did these enable that prior works could not?  Even a trend plot of papers over time, with some mentioning “method A works better than prior method B because of this one pivotal shift…”, would strengthen the evidence for this claim, which, in several parts, is unsubstantiated.

**Requested Changes:**

1. (Critical) Remove, tone down, or properly justify the claims 1-3 that i described above

    1. “Code constitutes the universal action space for multimodal general intelligence”

    2. “Visually-grounded code generation marks a definitive breakthrough toward autonomous software agents”

    3. “Four pivotal technical shifts”

2. (Strengthen) Add a methodology section or paragraph explaining how the survey was done, what tools were used, and the criteria for including works

3. (Critical) Utilize Section 2’s decomposition throughout the other sections, or provide a clear justification as to why it's not used. Right now, this formalism is introduced without being utilized in the manuscript. Alternatively, it would be good to categorize the works in a table with this (which work does what?) . This could even be in the (Anonymous) Github README.

---

> ### Author Response · Authors · 2026-06-07
> **Response to Reviewer Cqr5**
>
> We thank the reviewer for the detailed comments. We agree that the original manuscript does not make its analytical structure, claim support, and use of the Section 2 formalism sufficiently explicit. In the revision, we make a full-pass rewrite of the framing in Section 1 (Introduction), the formulation connections in Section 2 (Task Formulation), the domain openings and Takeaway boxes in Sections 3--6, Section 7 (Future Directions), Section 8 (Limitations), and Section 9 (Broader Impact), with all revised content marked in red.
>
> **Reducing overclaims.** We agree that the original Abstract overclaims the role of code. We remove the statements that "code constitutes the universal action space for multimodal general intelligence" and that visually grounded code generation is a "definitive breakthrough" toward autonomous agents. The revised Abstract now states a narrower conclusion: visual plausibility is an incomplete correctness signal, and reliable multimodal code intelligence requires evidence about artifact semantics, execution states, task transfer, and replayable agent traces.
>
> **Replacing "four pivotal shifts" with a verification-centered agenda.** We agree that the previous future-direction framing makes broad claims without enough support. We rewrite Section 7 (Future Directions) around the validation gaps identified in Sections 3-6. The four directions are now presented as an organizing agenda rather than as a claim that the field has uniformly shifted in these four ways: Section 7.1 discusses multi-signal validation, Section 7.2 discusses multi-state verification, Section 7.3 discusses cross-task transfer testing, and Section 7.4 discusses verifiable agent traces. This framing is connected to the Scope and Trajectory paragraphs and Takeaway boxes in the main technical sections.
>
> **Using the Section 2 formalism.** We agree that the original Section 2 (Task Formulation) taxonomy is underused. We revise the paper so that Section 2 is used more consistently as an analysis lens. Specifically, we revise the domain introductions in Section 3 (Graphical User Interface), Section 4 (Scientific Visualization), Section 5 (Structured Graphics), and Section 6 (Frontier Tasks and Frameworks), the taxonomy captions in Figure 4 and Figure 5, the Scope and Trajectory paragraphs, and multiple subsection openings to state more clearly whether the task is direct generation, editing, refinement, programmatic tool use, or executable policy generation, and what evidence each task treats as correctness.
>
> **Moving beyond a categorized bibliography.** We revise the section-level writing to emphasize bottlenecks and validation gaps in addition to paper coverage. For example, Section 3 (Graphical User Interface) now contrasts static visual reconstruction with dynamic runtime behavior. Section 4 (Scientific Visualization) separates rendered plausibility from data, structure, and scientific semantics. Section 5 (Structured Graphics) distinguishes visual similarity from symbolic editability and constraint validity. Section 6 (Frontier Tasks and Frameworks) focuses on process faithfulness, state trajectories, and agent traces. These revisions are intended to make the survey more analytical and less purely taxonomic.
>
> **Benchmark saturation, leakage, and metric reliability.** We add targeted discussion of benchmark reliability in the relevant sections and in Section 8 (Limitations). The revised text discusses synthetic template leakage, memorized renderer patterns, benchmark saturation, VLM-as-judge sensitivity, human preference score limitations, and the need for provenance and de-duplication checks. Table 8 in Section 7.1 also makes metric and judge failure modes more explicit.
>
> **Survey methodology.** We add a Survey Methodology paragraph in Section 1 (Introduction). It now summarizes the collection period, source types, inclusion/exclusion criteria, taxonomy assignment, and manual checks. This may address the concern that the paper uses the language of a systematic survey without explaining the review protocol.
>
> **Limitations and broader impact.** We add Section 8 (Limitations) and Section 9 (Broader Impact). Section 8 states the boundaries of public-paper coverage, closed-source systems, rapidly changing arXiv releases, incompatible metrics, and underexplored issues such as maintainability, security, accessibility, latency, user trust, and agent-trace auditing. Section 9 discusses productivity benefits, privacy risks, insecure or misleading generated code, and safety risks in agentic and embodied systems.

---

### Review · Reviewer_Dfvf · 2026-05-25

**Summary Of Contributions:**

This review is conducted using this guidelines from the TMLR FAQs.

> we want survey papers that draw new, previously unreported connections between several pieces of work in an area, and/or that clearly highlight trends in the area and/or suggest currently open problems.

This paper surveys “Multimodal Code Intelligence,” organizing previous work into GUI code generation, scientific visualization, structured graphics, and frontier frameworks such as programmatic visual manipulation, video-to-code, embodied control, and visually grounded programming. Its main contribution is a broad taxonomy of benchmarks and methods.

Trends highlighted:
* From SFT to RL training.
* Chat style to agentic workflow.
* Towards unified code model.

Open problems:
* Visual rewards for RL.

**Audience:**

Yes

**Audience Explanation:**

Multimodal code generation is an important topic and at the intersection of 2 large fields. I believe a well constructed survey would get the interest of the community.

**Broader Impact Concerns:**

Coding agents taking more and more of the SWE role is an important shift both societal and technical. Software might become more or less reliable, etc. and SWE roles will be changing. This paper does not address any of the broader impacts.

**Claims And Evidence:**

No

**Claims Explanation:**

Here is a list of claims that could use more supporting evidences.

* Claim: The paper is a systematic and comprehensive survey.

I think this claim needs more support. I would like to see the authors describe their search procedure, inclusion/exclusion criteria, cutoff date, screening process, and how the anonymous repository was curated. Without this, the paper reads more like a broad curated literature map than a systematic survey.

* Claim: Reliable visual rewards for RL is an open problem.

> “However, unlike conventional code generation, which can rely on deterministic unit testing for rewards, multimodal code generation lacks precise automated metrics for evaluating visual output. Consequently, **designing reliable reward functions and robust training strategies for RL remains a critical direction for future research.**”

Indeed, it looks like different works on different subfields of multimodal code generation use different reward. Perhaps a table describing what type of rewards are used and what they fail to capture in each situation would be helpful.

* Claim: community is moving away from sft to rl training for multimodal code generation.

> "A pivotal evolution in the Multimodal Code Intelligence landscape is the strategic transition from SFT to RL.”

Perhaps a chart showing the trend of RL papers in absolute number or proportionally to SFT papers would be beneficial. Additionally, I would like to see a table with the following columns problem, SFT SoTA and RL SoTA to see if indeed RL methods do perform better overall in multi modal code generation

**Requested Changes:**

* Add table describing what type of rewards are used and what they fail to capture in each situation.
* Add table with the following columns problem, SFT SoTA and RL SoTA
* Add chart of RL papers proportionally to SFT papers accepted at top conferences over time.
* Describe their search procedure, inclusion/exclusion criteria, cutoff date, screening process, and how the anonymous repository was curated
* Add a broader impact section.

---

> ### Author Response · Authors · 2026-06-07
> **Response to Reviewer Dfvf**
>
> We thank the reviewer for identifying the need for stronger evidence behind the claims of systematic coverage, visual rewards, and the SFT-to-RL trend. To address these concerns, we revise the full manuscript with red-marked changes, add survey methodology details in Section 1 (Introduction, Survey Methodology), and add Figure 1 and Figure 3 to quantify the publication trend and paper coverage of the main technical sections.
>
> **Survey methodology.** We agree that the original paper does not sufficiently explain how the literature is collected and screened. We revise the title/framing and add a dedicated Survey Methodology paragraph in Section 1 (Introduction) to make the review protocol explicit. This paragraph describes how papers are collected, screened, and assigned to the taxonomy, and states the January 2026 cutoff. It also defines the scope as work where visual inputs, visual outputs, or visually grounded states are tied to code generation, editing, verification, execution, or reasoning, while excluding purely text-only code-generation work unless visual evidence is part of the task or evaluation.
>
> **Coverage and trend evidence.** To make the coverage more concrete, we add Figure 1 and Figure 3. Figure 1 reports the quarterly publication trend from 2024Q1 to 2025Q4 and annotates representative works. Figure 3 reports domain and subdomain coverage for the main technical sections. The sunburst in Figure 3 is derived from the revised Section 3--6 bibliography and reports 282 de-duplicated papers covered in the main technical sections. The quarterly trend in Figure 1 uses first-public dates estimated from BibTeX metadata, arXiv identifiers, venue records, release records, and manual title checks when needed. This provides clearer evidence about the literature snapshot and temporal trend.
>
> **Visual rewards and what they fail to capture.** We agree that "reliable visual rewards" should not be discussed as a generic open problem without specifying what current rewards observe and miss. We therefore add Table 8 in Section 7. The table separates visual, text/code, preference, agent replay, and trace judges. For each proxy, it lists representative uses, judge failure modes, and companion checks. This makes explicit that visual similarity can hide wrong data, invalid structure, non-editability, broken interaction, or unfaithful traces.
>
> **SFT-to-RL claim.** We agree that the original wording makes the SFT-to-RL shift sound more uniform than the evidence supports. We therefore tone down the claim in the Abstract and in Section 7 (Future Directions). The revised text no longer asserts a field-wide "pivotal transition" from SFT to RL. Instead, Section 7.1 describes a narrower trend toward feedback-aware optimization, where RL-style optimization, preference learning, rendering-aware rewards, and task-specific feedback represent ways to incorporate execution, rendering, or preference feedback in particular task families.
>
> **On the requested SFT SoTA vs. RL SoTA table.** We agree that a direct SFT-vs-RL comparison would be valuable, especially within the same task family and on shared benchmarks. The current revision does not include this table because the evidence is still uneven across multimodal code-generation subfields. This is particularly clear in SVG and CAD, where many papers introduce their own datasets and evaluation protocols, so reported SOTA results are often not directly comparable across papers. Given the revision timeline, this version does not yet include a careful and fair SFT/RL SOTA table. We apologize for this limitation. As a partial response, we add Figure 1 and Figure 3 make the publication trend and paper coverage more explicit, and Table 8 summarizes representative reward and judge signals. We plan to further update the paper in the next version with new research works and a more systematic SFT/RL comparison.
>
> **On the requested RL-paper proportion chart.** We also agree that a chart of RL papers relative to SFT papers would be useful for substantiating the training-trend claim. The current revision does not include such a proportional RL/SFT chart because SFT and RL are difficult to count as binary categories in this field. In many works, SFT is the foundation used before RL, so a simple SFT-versus-RL proportion could misrepresent the training pipeline. We therefore tone down the claim and use Figure 1 to report the broader quarterly publication trend; we plan to add a more fine-grained SFT/RL analysis in the next update.
>
> **Broader impact.** We add Section 9 (Broader Impact) to discuss both benefits, such as lower barriers to visual programming, and risks, including incorrect or insecure code, privacy leakage, and unsafe agentic or embodied actions.